

# Phosphorus attenuation in streams by water-column geochemistry and benthic sediment reactive iron

Zachary P. Simpson[1], Richard W. McDowell[1,2], Leo M. Condron[1]

[1]Department of Soil and Physical Sciences, Lincoln University, Lincoln, 7647, New Zealand
[2]AgResearch, Lincoln Science Centre, Lincoln, 7647, New Zealand

*Correspondence to*: Zachary P. Simpson (zpsimpso@gmail.com)

**Abstract.** Streams can attenuate inputs of phosphorus (P) and, therefore the likelihood of ecosystem eutrophication. This attenuation is, however, poorly understood, particularly in reference to the geochemical mechanisms involved. In our study, we measured P attenuation mechanisms in the form of (1) mineral (co-)precipitation from the water-column and (2) P sorption
with benthic sediments. We hypothesized that both mechanisms would vary with catchment geology and, further, that P sorption would depend on reactive Fe content in sediments. We sampled 31 streams at baseflow conditions, covering a gradient of P inputs (via land use), hydrological characteristics, and catchment geologies. Geochemical equilibria in the water-column were measured and benthic sediments (<2 mm) were analyzed for sorption properties and P and iron (Fe) fractions. Neither P-containing minerals (e.g., hydroxylapatite) nor calcite-phosphate co-precipitation had the potential to occur. In contrast, in-
stream dissolved reactive P (DRP) correlated with labile sediment P (water-soluble and easily reduced Fe-P), but only for streams where hyporheic exchange between the water-column and the coarse sediment porewaters was likely sufficient. The non-labile P fractions contained most of sediment P (generally >90%) and varied with parent geology. Similarly, most sediment Fe was in a recalcitrant form (generally >90-95%). However, despite its small contribution to total sediment Fe, the pool of surface-reactive Fe was a strong predictor for sediment P sorption potential. Our results suggest that, in these streams, it is the
combination of biogeochemical Fe and P cycles and the exchange with the hyporheic zone that attenuates DRP in baseflow. Such combinations are likely to vary spatiotemporally within a catchment and must be considered alongside inputs of P and sediment if the P concentrations at baseflow – and eutrophication risk – are to be well managed.

## 1 Introduction

The mitigation of phosphorus (P) pollution is necessary to manage the eutrophication of water bodies. The efficacy of
mitigation measures is typically hoped for in stream monitoring data – a change within the catchment could decrease export of P downstream – but such effects are easily masked (Meals et al., 2010) due to the many biogeochemical processes integrated into the P signal (Dupas et al., 2018a; Halliday et al., 2012). Once mobilized from land via surface runoff or sub-surface flows (McDowell et al., 2004, 2015), P is repeatedly impeded along its flowpath by biotic and abiotic processes (Huang et al., 2018). This persistence of P gives rise to 'legacy P' (Powers et al., 2016; Sharpley et al., 2013), where past P inputs can take unknown



years (decades, centuries) to deplete. For example, McCrackin et al. (2018) estimated that only 6% of all anthropogenic P
inputs to the Baltic Sea drainage basin for 1900-2013 have discharged into the sea. Knowing the mechanisms behind legacy P
will improve predictions of P transport and thus not only engender wise P management but appropriate assessment of
management as well.

Legacy P refers to the attenuation – and slow trickle downstream – of P across the entire catchment, but perhaps the most

reactive flowpath for P is the stream itself (Ensign and Doyle, 2006; Haggard and Sharpley, 2007). At baseflow, P in streams
is subject to biotic and abiotic processes which could lead to the transient storage of P (within the channel or in biota) or its
re-mobilization back to the water column (Hall et al., 2013; House, 2003). Both periphyton (Biggs, 2000; Davies and Bothwell,
2012; Hill et al., 2009) and heterotrophic microbes (McDowell, 2003; Mulholland et al., 1997; Muscarella et al., 2014) can
assimilate P, especially dissolved reactive P (DRP; mostly orthophosphate but can include labile organic compounds), as it is

the most bioavailable P form (Biggs, 2000; Muscarella et al., 2014). The turnover of this assimilated P back to the water
column completes one cycle of the biotic P attenuation (or spiraling) mechanism (Ensign and Doyle, 2006; Newbold et al.,
1983). Numerous studies on strictly biotic P attenuation, however, conclude that biotic mechanisms are not critical for P
attenuation in streams (Griffiths and Johnson, 2018; Hall et al., 2002; Weigelhofer et al., 2018). Instead, abiotic P mechanisms
– as influenced by stream biogeochemistry – may be responsible for attenuating P (Jarvie et al., 2006; McDaniel et al., 2009;

Stutter et al., 2010). Yet, abiotic (i.e., geochemical) P attenuation phenomena are poorly studied relative to biotic P attenuation.
To address this gap, we consider two geochemical mechanisms – calcium (Ca) based (co-)precipitation and sediment P sorption
– as the primary components of abiotic P attenuation in streams.

Calcium-phosphate mineral precipitation and $CaCO_3$ co-precipitation may remove DRP from the water column given sufficient
Ca, pH, and $p\mathrm{CO2}$ (Golterman, 2004; House, 2003; Stumm and Morgan, 1996). This may contribute to the initial removal of

DRP from the water column, whereby further adsorption or mineral transformations (e.g., towards hydroxyapatite) may occur
(Diaz et al., 1994; Golterman, 1988; Plant and House, 2002). Given the dependence on water-column geochemistry, Ca-based
P attenuation is likely a function of catchment geology (Corman et al., 2015; House, 2003; Mulholland et al., 1997). Indeed,
most studies focused on Ca-based P attenuation are largely located in catchments with calcareous (i.e., chalk, karst) geology
(Cohen et al., 2013; Diaz et al., 1994; House, 1999; Jarvie et al., 2006). Few studies have considered a range of geologies

where other abiotic P attenuation mechanisms may diminish the importance of Ca-based P attenuation, particularly sediment
P sorption.

Benthic stream sediments can have large potential to adsorb P and thus attenuate P inputs (Froelich, 1988; Haggard and
Sharpley, 2007; McDowell, 2015), especially for baseflow conditions where water is given time to contact sediments in the
hyporheic zone (Harvey, 2016). Stream sediments provide abundant reactive surfaces for P sorption; notably, clay minerals

and metal (i.e., iron and aluminum) oxides are strong reaction sites (Gérard, 2016; Golterman, 2004; Parfitt, 1979). Further,
amorphous iron (Fe) oxides, a variable fraction of the total sediment Fe pool (Jan et al., 2013; Parsons et al., 2017), have great
affinity for P (Goldberg and Sposito, 1984; Lijklema, 1980) and are the primary reaction sites in many non-calcareous streams
(Dupas et al., 2018b; van der Grift et al., 2014; Lewandowski and Nützmann, 2010). This sediment sorption can be readily





examined via intensity of adsorption and the quantity of P already complexed with the sediment. Sorption intensity

measurements often correlate negatively with in-stream DRP (McDaniel et al., 2009; McDowell, 2015; Weigelhofer et al.,

2018) and positively with stream P uptake metrics (Demars, 2008; Haggard et al., 2005; Jarvie et al., 2005). Concordantly,

sediments in streams with high P loading (i.e., high sustained DRP concentrations) tend to have diminished sorption ability

and greater stores of P (Jarvie et al., 2012; McDowell, 2015), particularly in the more labile and redox-sensitive pools

(Lewandowski and Nützmann, 2010).

Therefore, in the present study, we examined P attenuation mechanisms in streams at baseflow via Ca-P geochemistry in the

water column, stores of sediment P and Fe, and P sorption capacities of sediments. Given that these processes are likely all

tied to catchment geology and P inputs, we sampled waters and benthic sediments of streams in the Canterbury region, New

Zealand, covering a variety of geologies, land use, and stream characteristics. We hypothesized that the primary mechanisms

responsible for P attenuation (and therefore related to DRP concentrations) were Ca-based mineral equilibria in the water

column and sorption with benthic sediments. Further, we hypothesized that amorphous, reactive Fe was a primary controller

of sediment P sorption, rather than refractory or total Fe pools.

## 2 Materials and methods

### 2.1 Study sites

We sampled 31 streams in the Canterbury region, New Zealand (Fig. 1). The site characteristics are given in Table S1,

according to the River Environment Classification (REC) developed for New Zealand (Snelder and Biggs, 2002). Stream sizes

were mostly 1st to 5th order, with some 6th and 7th order streams (*n*=7), and generally included both low-elevation and

hilly/mountainous streams. Most of the catchments contained some amount of pastoral land uses (i.e., sheep and dairy farming),

reflecting the dominant land use in the Canterbury region. In north Canterbury, basins are characterized by quaternary fluvial

gravels with some underlying sedimentary deposits (e.g., limestone); further south, the plains were formed by river-deposited

erosion and glacial outwash products (by the Rakaia and Waimakariri Rivers), with some intermittent outcrops of greywacke;

Banks Peninsula is characterized by a basalt volcanic geology (Brown, 2001). Using the scheme provided by the REC,

catchment geology of the study streams corresponded to alluvium (*n*=15), sedimentary (hard and soft sedimentary; *n*=10), and

volcanic basic (*n*=6); for presentation and discussion purposes, we grouped the observed data in this paper based on these

geology classes (see below). Further, we used the REC to distinguish between two prominent sources of flow for the streams

by identifying spring-fed sites (*n* = 8) from the other sites (simply termed here as 'hill-fed'; *n* = 23).

### 2.2 Sampling procedure

Sediment and stream water samples were collected at each site during baseflow conditions from March to May, 2018 (austral

late summer/early autumn). We sampled between 1000 and 1600 h to minimize possible diel effects on DRP (Cohen et al.,

2013; McDowell et al., 2019). Benthic sediments (top 1-3 cm) were collected in-*situ* with a scoop and wet-sieved in the field





to <2 mm. The sieved sediment slurry settled after 30 minutes, where excess water was decanted and ~2 kg of wet sediment was stored on ice and later refrigerated in the laboratory (4 °C). Sediment sampling locations were targeted within the stream where surface water interacts with benthic sediments; primarily, riffle beds near the centroid of flow were sampled but depositional areas closer to the bank were sampled at sites where the stream was too deep and fast-flowing for practical sampling.

Water grab samples were collected from the centroid of flow via wading (approaching site from downstream) or with an extended pole and bottle. Sample bottles were field-rinsed three times before taking a sample at two-thirds of the stream depth. Two subsamples were filtered in the field (0.45 µm) while another subsample was left unfiltered (all with minimal headspace); all samples were then stored on ice for transport back to the laboratory followed by either freezing (-20 °C; for ion chromatography, ICP-OES, and dissolved Fe as explained below) or refrigeration (4 °C). In addition, we measured dissolved
oxygen and temperature at each stream.

## 2.3 Water physicochemical analyses

Upon return to the laboratory, we immediately measured pH and conductivity in the unfiltered stream sample. Alkalinity was measured on the filtered sample within 24 h of collection via Gran's titration method (Gran, 1952), following the protocol of Rounds (2012) with 0.05 M HCl as the titrant. We also measured DRP on the filtered sample within 24 h via the malachite-
green method (detection limit of 0.006 mg P L-1; Ohno and Zibilske, 1991; D'Angelo et al., 2001) to minimize storage influences (Jarvie et al., 2002). Other analyses on the filtered sample included dissolved anions (F, Cl, SO$_4$, NO$_3$) via ion chromatography (detection limits range 0.02 to 0.50 mg L$^{-1}$), cations (Al, Ca, Fe, K, Mg, Mn, Na, Zn) via ICP-OES (detection limits approximately 0.002 mg L$^{-1}$), and total dissolved Fe via a ferrozine method (see below); other elements (e.g., some trace metals) were below detection. Total suspended solids (TSS; method 2540D (APHA, 2005)) were low in these streams at
baseflow (mean of 7 mg L$^{-1}$), so suspended sediment likely had negligible influence on DRP (data not shown). Blanks and duplicate checks were included in each batch of samples to ensure quality of results.

We modified the ferrozine colorimetric method (Stookey, 1970) developed by Viollier et al. (2000) to measure total dissolved Fe in stream samples and P fractions (below), as it was cost-effective and provided greater sensitivity than ICP-OES for concentrations <0.050 mg Fe L$^{-1}$. Full details are in the supplementary material (S1.3). Briefly, the aliquot reacted with the
ferrozine reagent and the reducing agent for 16 h under light conditions (to benefit from photoreduction; Anastácio et al., 2008) before being buffered at pH 9.5 and reading the absorbance at 562 nm. The method detection limit with a 1 cm light path was approximately 0.010 mg Fe L$^{-1}$ in solutions and 0.017 mg Fe L$^{-1}$ for digests.

## 2.4 Sediment physicochemical analyses

A sediment sample was dried at 104 °C to measure moisture content. Fresh sediment pH was measured in D.I. water at 1:5
sediment to solution ratio. We measured two common P sorption indices on fresh sediment: anion storage capacity (ASC; Blakemore et al., 1987) and the Bache-Williams index (BWI; Bache and Williams, 1971) as modified by Burkitt et al. (2002).



Both are single-point isotherms with overnight shaking (16 h) but differ in their conditions. ASC uses solution that is buffered at pH of 4.6 with 1000 mg P L$^{-1}$ (1:5 sediment to solution ratio) and expressed as percent removal of P. BWI uses a 0.01 M CaCl$_2$ solution (no pH buffer) with 1000 mg P kg$^{-1}$ sediment added (here, 100 mg P L$^{-1}$ at sediment to solution ratio of 1:10)

and is expressed as P sorption (mg P kg$^{-1}$ sediment) divided by $\log_{10}(c_e)$ where $c_e$ is the remaining equilibrium solution concentration in µg P L$^{-1}$. After centrifuging (2400 $g$ for 15 minutes), the supernatants were analyzed for DRP via the molybdenum-blue method (Murphy and Riley, 1962).

A subsample of sediment was freeze-dried and analyzed for total C and N with a CN elemental analyzer and for total element concentrations with ICP-OES following microwave digestion with nitric acid plus hydrogen peroxide. Additionally, dried

sediment samples were sieved to <1 mm and analyzed for particle size distributions via laser-diffraction, which are expressed on a percent volume basis (Eshel et al., 2004); here, we define clays as ≤ 4 µm, silt as > 4 µm and ≤ 62.5 µm, and sands as > 62.5 µm.

## 2.5 Sediment phosphorus fractionation

We began sediment P fractionation within a week of sampling using fresh (not dried) sediments (Simpson et al., 2019)

following the scheme of Jan et al. (2015). The primary details of the sequential fractionation are given here and in Table 1; further details are given in the supplementary material (S1). We used 0.5 g d.w. sediment and 10 mL of extractant in each step, in triplicate. The bicarbonate-dithionite (BD) solution (0.1 M NaHCO$_3$ and 0.1 M Na$_2$S$_2$O$_4$, pH 7.2; BD-I and BD-II fractions) was prepared fresh with de-aerated D.I. water (subject to vacuum for 30 min) and used immediately. The NaOH (I and II) and HCl fractions used 1 M NaOH and 0.5 M HCl, respectively. A 0.5 M NaCl wash step was included after the BD-II and NaOH-

II steps to prevent carryover to the next fraction (Condron and Newman, 2011; Jensen and Thamdrup, 1993). Following the extraction at room temperature with an end-over-end shaker, we centrifuged (10 min at 2400 $g$) and filtered the extracts (Whatman grade 41). The BD-I and NaOH-I fractions involved 5 min of shaking, immediate centrifuging, decanting, and a further 5 min extraction (i.e., 10 minute total extraction time) as per Jan et al. (2015).

We did not analyze DRP in the BD extracts due to analytical difficulty (Lukkari et al., 2007); however, we suspected little, if

any, organic P would contribute to the TP signal since BD primarily targets phosphate associated with reducible metal oxides (Jan et al., 2013; Jensen and Thamdrup, 1993). Although we used a colorimetric method suited for alkaline extracts (He and Honeycutt, 2005), RP concentrations in the NaOH fractions were likely compromised by silicates since the strong alkaline solution solubilizes silica minerals (Lindsay, 1979; Sauer et al., 2006) and silicate produces a molybdenum-blue complex analogous to phosphate (Nagul et al., 2015). However, NaOH-TP values are still valid since digestion removes the silicate

interference (Malá and Lagová, 2014; Zhang et al., 1999). Therefore, we restrict our discussion to NaOH-TP. Total P in the BD and NaOH fractions was determined via acid-persulfate autoclave digestion (method 4500-P; APHA, 2005) followed by the molybdenum-blue method (method detection limit of ~0.02 mg P L$^{-1}$ for digests). More analytical details are in the supplementary material (S1.2).



In addition to the P in each fraction, we measured total Fe in the BD and NaOH digests with the ferrozine method described
above. We examined patterns in Fe and P contents among the BD and NaOH fractions with molar Fe:P ratios. All fractionation
data presented in this paper are based on the averages of the laboratory triplicate analyses.

## 2.6 Geochemical equilibria

We examined phosphate mineral equilibria with regard to streamwater chemistry to determine the direction of potential
precipitation/dissolution reactions (Pierzynski et al., 2005). We employed the PHREEQC geochemical model (Parkhurst and
Appelo, 2013) with the MINTEQA2 version 4 database to calculate mineral saturation indices (SIs) and therefore discuss
saturation states with reference to the minerals (Appelo and Postma, 2005). A mineral's SI is defined as $\log_{10}(IAP/K_{sp})$, where
IAP is the ion activity product measured in solution and $K_{sp}$ is the mineral's equilibrium solubility constant. A SI >0 and SI
<0 indicate super- and subsaturation with respect to the mineral phase. We employ these data to detect if the thermodynamic
equilibria favor precipitation reactions as a potential mechanism for DRP removal, but cannot determine what phases actually
occur as there may be kinetic limitations (Plant and House, 2002; Stumm and Morgan, 1996). The analytical input data
consisted of: stream temperature, pH, total dissolved anions and cations, alkalinity, and DRP. Here, we assumed DRP to be
primarily orthophosphate. Since these streams were above or near oxygen saturation at time of sampling, we assumed that Fe
was in an oxidized state (Fe(III)).

## 2.7 Data and statistical methods

Two sites in the sedimentary class – both at pristine, forested headwaters – had DRP concentrations below our detection limits.
To mitigate potential bias, we inserted reference DRP values based on previous analyses for streams under the same
classification within the REC (McDowell et al., 2013). Additionally, through exploratory analyses, we found it necessary to
remove data with a spring-fed source of flow as a confounding variable when modelling DRP (see below). We hypothesized
that the spring-fed streams had greatly diminished hyporheic exchange – thus limiting the interaction between sediment
reaction sites and the water column. This limitation could be due to 1) accumulation of fine, silty sediments which restricted
hydraulic conductivity (Aubeneau et al., 2014; Packman and Salehin, 2003; Weigelhofer et al., 2018), 2) low-gradients which
limited hydrodynamic forces at the streambed (Boano et al., 2014), and 3) the likely presence of groundwater inputs along the
reach, which are known to limit hyporheic exchange flows (Azizian et al., 2017).

We summarized differences between the stream and sediment variables (physico- and geochemical) between the three geology
classes with nonparametric tests (Hollander et al., 2013). We first tested the null hypothesis ($H_0$) that the locations of the group-
wise distributions were equal via the Kruskal-Wallis test; if $H_0$ was rejected, we then constructed multiple comparisons with
rank statistics, adjusting for simultaneous inferences (Konietschke et al., 2012, 2015). Additionally, Spearman correlations
were used to describe relationships between variables of interest. We explored the sediment-P attenuation mechanism in
streams by fitting simple predictive models (Shmueli, 2010); we used robust linear models ("rlm" function in MASS package;
Venables and Ripley, 2002) to model DRP and sediment P sorption (see supplementary material; S2). All tests were performed





at 95% confidence and all analyses were conducted in R (version 3.5.2; R Core Team, 2018). The data can be found at Figshare (https://figshare.com/s/718226c7f1940d631755).

## 3 Results

### 3.1 Stream water chemistry and mineral equilibria

Stream physicochemistry at the time of sampling is summarized in Table 2. The streams ranged from low to moderate alkalinity (16.7 to 88.8 mg $CaCO_3$ $L^{-1}$, mean of 42.7 mg $CaCO_3$ $L^{-1}$) and mean conductivity was 143 µS $cm^{-1}$. For the time of sampling (generally, 1000 to 1600 h), pH averaged 7.44 (6.43 to 7.93). Dissolved reactive P varied among geology classes with the lowest median DRP concentration in sedimentary, followed by alluvium, and then by volcanic basic (5.0, 7.4, and 27.7 µg P $L^{-1}$, respectively).

Stream geochemical equilibria modelling with PHREEQC was used to examine important mineral phases in the water column (Fig. 2). Stable, yet kinetically slow-to-form, Al and Fe minerals such as gibbsite and goethite were supersaturated in the water column; amorphous Al species ($Al(OH)_3(am)$) were subsaturated. The most reactive Fe species studied here, ferrihydrite ($Fe(OH)_3(am)$), was supersaturated (overall mean SI $1.9 \pm 0.6$), particularly over volcanic basic geologies (mean SI $2.9 \pm 0.2$). We note that ferrihydrite SI correlated with log-activity of $HPO_4^{2-}$ ($\rho = 0.71$, $p < 0.001$). Further, $CaCO_3$ precipitation was not

favorable (SI range of -2.1 to -0.08). Therefore, for these grab samples, $CaCO_3$ co-precipitation of phosphate was not a likely mechanism.

    Phosphate mineral solubilities were also investigated. The activities of phosphate and Fe/Al were too low to precipitate any P minerals from the water column in these streams (e.g., strengite; Fig. 2). For Ca-phosphate minerals, the less thermodynamically stable phases (e.g., $CaHPO_4 \cdot (H_2O)_2$) were sub-saturated in all samples. However, several streams showed

saturation to slight super-saturation with respect to hydroxylapatite; log-activity of $HPO_4^{2-}$ is plotted against a function of $Ca^{2+}$ and $H^+$ activities with reference to the standard solubility curve of hydroxylapatite in Fig. 3. All three geologies had some waters with positive hydroxylapatite SIs but, notably, the saturation appears to not extend significantly past the hydroxylapatite solubility curve (max SI of 1.7). We noted no other geochemically significant relationships for other mineral SIs (Fig. S1) or ion log-activities (Fig. S2). Given that most phosphorous minerals of interest were sub-saturated in the water column, and that

hydroxylapatite likely requires greater SI values before actively precipitating (see discussion below), mineral precipitation does not seem a significant mechanism for P removal in these streams.

### 3.2 General stream sediment physicochemistry

    Physicochemical characteristics of the benthic sediments are summarized in Table 3. The sediments were largely neutral (mean pH of 7.10). The fine sediments sampled in this survey were predominantly sandy (mean sand content of 84%), although five

alluvium sites and two volcanic basic sites had less than 80% sand content (silt + clay content range from 21 to 92%). Sediments from spring-fed streams were much finer ($D_{50}$ of $171 \pm 172$ µm) than those in hill-fed streams ($597 \pm 172$ µm). The sediments





were relatively low in total C (overall median of 3.0 g C kg$^{-1}$) except for the spring sites (median of 20 g C kg$^{-1}$). Owing to the different geological origins, the volcanic basic sediments were enriched with Al, Fe, Mn, and P in comparison to the sedimentary and alluvium sediments.

### 3.3 Stream sediment phosphorus and iron fractionation

#### 3.3.1 Phosphorus fractionation

Sediment P concentration fractionated according to the procedure by Jan et al. (2015) varied considerably by the catchment geology (Fig. 4). The most labile pool, H$_2$O-P, was relatively high in the volcanic basic sediments (mean ± SD; 3.84 ± 3.76 mg P kg$^{-1}$) and alluvium sediments (1.82 ± 1.41 mg P kg$^{-1}$), but lower for sedimentary sites (0.44 ± 0.08 mg P kg$^{-1}$; Table 4). Both reductively-soluble sediment P pools, BD-I and BD-II, were enriched in alluvium and volcanic basic sediments in comparison to sedimentary sites, with total BD-extractable P (BD-I plus BD-II) of 84.3 ± 84.1, 77.6 ± 26.7, and 6.60 ± 3.03 mg P kg$^{-1}$, respectively. The mean proportion of the reductively-soluble sediment P pool attributed to amorphous Fe oxides (BD-I), rather than crystalline Fe oxides (BD-II), was 62% in alluvium sediments but evenly distributed in volcanic basic sediments (49%). Both BD-I and BD-II P fractions correlated with fines (clays plus silts) concentration (respectively, $\rho = 0.89$ and 0.84, $p < 0.001$ for both tests; Fig. S3). Further, the labile P pools of H$_2$O-P and BD-I P together comprised only 0.5 to 21% of total P, with means of 1.0 ± 0.5, 8.2 ± 6.7, and 2.1 ± 1.4% for the sedimentary, alluvium, and volcanic basic geologies, respectively.

The NaOH-extractable P pool averaged 34.3 ± 10.7, 91.8 ± 84.7, and 518 ± 329 mg P kg$^{-1}$ for sediments from the sedimentary, alluvium, and volcanic basic geologies, respectively. The more labile NaOH fraction (NaOH-I) made up the majority of the total NaOH-extractable P pool (67-71% on average). Total NaOH-extractable P correlated with Al content ($\rho = 0.47$, $p = 0.008$), but no other relationships (e.g., with total C) were evident. The least available sediment P pool analyzed, HCl-P, averaged 452 (± 161), 337 (± 163), and 1373 (± 572) mg P kg$^{-1}$ for sites in the sedimentary, alluvium, and volcanic basic geologies, respectively. As expected, HCl-P correlated strongly with Ca ($\rho = 0.84$, $p < 0.001$) and Mg content ($\rho = 0.71$, $p < 0.001$) since this fraction is dominated by primary mineral P.

In-stream DRP correlated well with both H$_2$O-P and the sum of H$_2$O-P and BD-I P (hereafter, labile P; Fig. 5). Using sediment P pools as the primary predictors for DRP in hill-fed streams (Table S2), the best-fit linear model employed geology and H$_2$O-P (RMSE = 3.93 µg P L$^{-1}$), with a slope (95% C.I.) of 2.62 (1.58 to 3.66) µg P L$^{-1}$ per mg P kg$^{-1}$ for H$_2$O-P.

#### 3.3.2 Iron fractionation

In addition to P, we measured Fe in selected extracts of the P fractionation scheme (Fig. 6). Although total Fe is similar between alluvium and sedimentary sites (means of 22.0 and 23.4 g Fe kg$^{-1}$; Table 3), reactive Fe pools varied markedly within and between all catchment geologies (Table 4). Total BD-extractable Fe was 2280 (± 2200), 700 (± 242), and 7710 (± 2120) mg Fe kg$^{-1}$ for the alluvium, sedimentary, and volcanic basic sites, respectively. On average, the amorphous Fe pool made up 52,





38, and 34% of the BD-extractable Fe in alluvium, sedimentary, and volcanic basic sediments. The BD-extractable pools of Fe correlated with percent fines (Fig. S4), where Spearman $\rho$ was 0.87 ($p < 0.001$) and 0.79 ($p < 0.001$) for BD-I and BD-II,

respectively. The Fe extracted by NaOH was at least one order of magnitude less than BD-extractable Fe for each geology (totals of 59 to 250 mg Fe kg$^{-1}$); NaOH-I Fe related to percent fines ($\rho = 0.77$, $p < 0.001$) and total C ($\rho = 0.75$, $p < 0.001$), which is likely due to release of Fe complexed with organic matter during extraction. In general, the BD- and NaOH-extractable Fe scaled with total Fe ($\rho = 0.45$ ($p = 0.012$) and $\rho = 0.39$ ($p = 0.033$), respectively). However, reactive Fe contributed only a minor portion of total Fe in these sediments ($8.6 \pm 7.2$% and $0.6 \pm 0.8$% for total BD- and NaOH-extractable Fe, respectively;

Fig. 6).

### 3.3.3 Fe:P ratios in sediments

Molar Fe:P in pools of sediment P varied depending on the pool and the catchment geology (Table 4, Fig. S5). In general, Fe:P ratios were much lower for the NaOH fractions (approximately 0.25 to 3 mol Fe mol P$^{-1}$) than for other fractions (>13 mol Fe mol P$^{-1}$), likely due to little surface reactive Fe left after BD extraction while containing P bound with constituents other than

Fe. A paired Wilcoxon signed-rank test on Fe:P ratios in the BD fractions indicated that BD-II Fe:P is, on average, 25.6 (95% C.I., 14.4 to 36.1) mol Fe mol P$^{-1}$ greater than BD-I Fe:P. However, Fe:P ratios for either BD-I or BD-II fractions did not improve linear model fits for either DRP or ASC (data not shown). For the BD (I and II) fractions, median Fe:P was less in alluvium sediments (15.5 and 19.9 mol Fe mol P$^{-1}$) compared to both sedimentary (36.4 and 80.7 mol Fe mol P$^{-1}$) and volcanic-basic sediments (39.1 and 85.8 mol Fe mol P$^{-1}$).

## 3.4 Stream sediment phosphorus sorption

Sediment P sorption metrics are summarized in Table 4. Although ASC (a measure of P retention at low pH) and BWI (P retention at neutral pH and controlled Ca concentration) differ in how P sorption potential is determined, a similar pattern was apparent in both variables for the three sampled geologies. Given the widespread use of ASC in soil classification (and management) in New Zealand (Saunders, 1965), we focus on ASC for brevity. The sedimentary samples had lower ASC (10.2

$\pm 5.6$ %) than either alluvium ($33.6 \pm 26.9$ %) or volcanic basic sediments ($49.4 \pm 11.1$ %). There was no clear relationship between ASC and sediment Fe:P ratios (Fig. S6), while ASC was correlated with BD-I ($\rho = 0.77$, $p < 0.001$), BD-II ($\rho = 0.65$, $p < 0.001$), and total Fe ($\rho = 0.55$, $p = 0.002$) pools (Fig. 7).

More refractory pools of Fe were less predictive of ASC than BD-I Fe (Fig. 7). While ASC for the refractory Fe pools tended to cluster according to geology and stream source of flow, ASC varied linearly as a function of BD-I Fe (all data), which we

illustrate with a linear modelling exercise, summarized in Table S3. The simple model of only BD-I Fe had the lowest AIC and the lowest RMSE (9.2 %); the estimated slope for BD-I Fe was 0.0174 (95% C.I., 0.0152 to 0.0197) % per mg Fe kg$^{-1}$. Additionally, while adding catchment geology did not improve the model fit for BD-I Fe, both the BD-II and total Fe models included this term for their respective optimal models. Thus, BD-I Fe alone predicted ASC optimally and captured the variance otherwise provided by geology or refractory Fe to the lesser models.





Contrary to our expectations, sediments with greater sorption potential positively correlated with in-stream DRP (Fig. 8), but only for the case of hill-fed streams. In addition to the linear models fitted with the labile P fractions above, we used ASC as a predictor of DRP (Table S2) – again, we excluded spring-fed sites as a confounding variable. Using ASC alone gave similar performance (RMSE of 4.44 µg P L$^{-1}$) as the models with terms for geology and the labile P fractions, but with less model degrees of freedom. Overall, the best linear model for DRP employed ASC and H$_2$O-P (RMSE of 4.38 µg P L$^{-1}$), with slopes

of 2.03 (1.22 to 2.83) µg P L$^{-1}$ per mg H$_2$O-P kg$^{-1}$ and 0.325 (0.218 to 0.431) µg P L$^{-1}$ per ASC %.

## 4 Discussion

Phosphorus in streams does not travel as a conservative solute. Our data suggests that bioavailable P, i.e., DRP and labile sediment P fractions, in these streams is likely to be modified by two principal factors: 1) reactive Fe content as derived from both geology and in-stream Fe cycling, and 2) the exchange between sediment porewaters and the water column (i.e., hyporheic

exchange) as governed by hydraulic forces and sediment particle size distribution. We summarize our lines of evidence to support this hypothesis below.

### 4.1 Stream water-column geochemistry and dissolved reactive phosphorus

Catchment geology is a primary control on the geochemical composition of stream water (Bluth and Kump, 1994; Stumm and Morgan, 1996), and our results suggest that the Ca-based mechanisms for P removal were insignificant in our streams. Among

the geochemical equilibria affecting phosphate in solution, phosphate-minerals rarely had the thermodynamic potential to form in these waters. The most stable phosphate mineral, hydroxylapatite, showed some cases of near-saturation (SI ≈ 0; Fig. 3). However, empirical research has suggested that the required supersaturation for hydroxylapatite to significantly precipitate from solution is much higher (reported SI from 3 to 10; House, 1999; Plant and House, 2002; Sø et al., 2011). Therefore, hydroxylapatite and other phosphate-mineral precipitation seems, at best, a secondary mechanism for P removal in these

streams. In contrast, streams with greater Ca concentrations (e.g., >100 mg Ca L$^{-1}$) and pH (>8) will likely be more conducive for Ca-P mineral precipitation (Diaz et al., 1994). It is striking that some streams approach, but do not significantly extend beyond, the hydroxylapatite solubility curve, which could suggest a role for hydroxylapatite equilibrium. However, stream solution chemistry is more complex than ideal solutions modelled by PHREEQC, where even the solubility constant for hydroxylapatite is subject to considerable uncertainty (Golterman, 2004). In addition, other phosphate-minerals (e.g., various

Al and Fe based phosphate-minerals) are unlikely to contribute to phosphate activity in the water column, but may be more important in some subsurface environments (e.g., Denver et al., 2010; Rothe et al., 2014).

Since, calcite co-precipitates with phosphate (Golterman, 2004; House, 2003), periods of calcite precipitation in streams may provide an opportunity for phosphate removal, particularly in low-phosphate systems (Machesky et al., 2010; Plant and House, 2002; Sø et al., 2011). House (1999) suggested that calcite precipitation in streams becomes significant near SI ≈ 1. In the

present study, however, we only observed negative SI's for calcite (Fig. 2). Our study design was more likely to capture the





upper extent of calcite SI variability since: 1) calcite saturation peaks during the day when photosynthesis depletes $CO_2(g)$ and increases pH (Nimick et al., 2011; Stets et al., 2017) and 2) calcite solubility decreases with greater temperatures during the day (Stumm and Morgan, 1996). It is likely that our study streams were not alkaline enough for significant calcite interactions (here, median alkalinities of 38.2 to 42.1, maximum of 88.8 mg $L^{-1}$ as $CaCO_3$; maximum pH of 7.93; Table 2) since streams

that reach a calcite SI $\geq$ 1 typically have alkalinity >100 mg $L^{-1}$ as $CaCO_3$ and sustain pH > 8 (Corman et al., 2015; Neal et al., 2002; Nimick et al., 2011; Stets et al., 2017). Hence, we rule out calcite-phosphate co-precipitation as a significant mechanism for DRP removal in these streams.

The removal of P in streams via Ca-P mechanisms is frequently documented (Burns et al., 2015; Cohen et al., 2013; Jarvie et al., 2006), but such studies are located in catchments dominated by carbonaceous/karst geologies. Since many streams suffering

from P pollution may not have such 'self-cleansing' mechanisms available, knowledge of other processes for P attenuation are critical for understanding P transport.

Aside from Ca-based solution geochemistry, an unexpected observation from this study was the supersaturation of ferrihydrite and its relationship with $HPO_4^{2-}$ activity (Fig. S1). Ferric iron is largely insoluble in most stream conditions (oxygenated water and pH near or above neutral) and Fe(II) would presumably be associated only with reducing zones within the stream corridor

(Stumm and Sulzberger, 1992), thus positive ferrihydrite SI seems implausible. Fox (1988) explained the problem that, in most streams, dissolved Fe is overestimated because common filtration methods (i.e., 0.45 μm filters) fail to remove colloidal Fe species (Baken et al., 2016b; van der Grift et al., 2014). Our apparent supersaturation with respect to ferrihydrite is within the range of previously observed over-estimates (SI up to 5 in most cases; Fox, 1988) and is a limitation of our dataset. However, the positive relation between ferrihydrite SI and DRP may indirectly point towards evidence of colloidal Fe species carrying

sorbed phosphate and should be investigated in future research.

## 4.2 Amount and form of sediment phosphorus depends on geology

Given the enormous amounts of sediment that stream networks retain and the long timescales for its transit (Czuba et al., 2017; Wohl, 2015), P bound in benthic sediments represents a crucial component of legacy P. However, we found that most sediment P was unlikely to be bioavailable in our study streams (Fig. 4). Taking $H_2O$ and BD-I extractable P (labile P) as potentially

bioavailable in lotic systems and definitely available in receiving lentic systems (e.g., lakes; Golterman, 2004; Jan et al., 2015; Jensen and Thamdrup, 1993; Monbet et al., 2010), sediment labile P was generally <10% of total P (maximum of 21% of total P), but varied with geology. Although small, these pools are highly reactive under baseflow and may be subjected to microbial turnover (McDowell, 2003), exchange with porewaters via desorption or reductive dissolution (Lewandowski and Nützmann, 2010; Parsons et al., 2017; Zak et al., 2006), and potentially to P-scavenging periphyton mats (Wood et al., 2015).

## 4.3 Amorphous iron content determines sediment phosphorus sorption capacity

Benthic sediment Fe has long been thought to provide much of the reactivity for P in streams (Danen-Louwerse et al., 1993; Froelich, 1988; Wauchope and McDowell, 1984). Therefore, we selected a fractionation scheme to target the responsible





fractions of Fe. To our knowledge, our study is the first application of the sediment P fractionation scheme developed by Jan et al. (2015) to stream sediments. The motivation behind this fractionation scheme is that, in the absence of precipitation

mechanisms, the more soluble, amorphous forms of Fe and Al oxides are the ones responsible for P retention and possible release (Jan et al., 2013, 2015; Postma, 1993). With this fractionation, we found large differences between catchment geologies and substrate texture (coarse sediments in hill-fed streams vs. silty sediments in spring-fed streams) in the amounts of reactive Fe (both BD-I and BD-II Fe) and its associated P, while total Fe was much less variable (Fig. 4 and Fig. 6). Further, BD-I Fe alone predicted sediment P sorption potential, as opposed to the more crystalline (BD-II) or refractory (total) Fe pools (Fig. 7,

Table S3). Hence, we focus our discussion around BD-I Fe and its implications for stream P transport.

The first BD step efficiently targets surface-reactive, amorphous Fe oxides, while the second BD step extracts more-crystalline, less reactive Fe oxides (Jan et al., 2015). This geochemical difference was evident in the Fe:P ratios (Table 4), where mean BD-I Fe:P ratios (varying with geology from $18.3 \pm 9.9$ to $65.0 \pm 66.9$) were significantly lower than BD-II Fe:P ($31.6 \pm 25.1$ to $99.2 \pm 61$), owing to the greater affinity for P in the amorphous fraction. Pure Fe-P minerals would have much lower Fe:P

ratios (e.g., 1 for $FePO_4$); laboratory studies have observed molar Fe:P ratios of approximately 2 to 5 for P incorporated into Fe oxides during precipitation (Deng and Stumm, 1994; Senn et al., 2015) or adsorbed onto fresh Fe oxide precipitants (Lijklema, 1980). In their study, Senn et al. (2015) report that Fe(III) precipitants in environmental conditions similar to that within many streams and hyporheic zones (circumneutral pH, varying amounts of Ca, $PO_4$, and silicate) are a complex mixture of basic (Ca-)$FePO_4$, ferrihydrite-type, and poorly crystalline lepidocrocite/goethite precipitants. This mixture of Fe(III)

precipitants with varying stoichiometries and phosphate sorption capacities may explain the wide variance we observed in the BD-I Fe:P ratios. For the BD-II fraction, Fe:P may be related to the weathering status of the parent sediment material (Lair et al., 2009), as more crystalline Fe oxides become less sorptive for P. Total Fe:P is less meaningful (Jan et al., 2013), due to the varying contributions of Fe oxides to the total Fe and of other pools of P (e.g., NaOH-extractable P) to the total P (Hoffman et al., 2009; Machesky et al., 2010). Here, the contribution of BD-extractable Fe to the total Fe was generally <10%, indicating

that much sediment Fe can be unimportant to in-stream P cycling.

While the Fe:P ratios differentiated pools of Fe according to their geochemical affinity for P (BD-I vs. BD-II), Fe:P ratios were poor predictors of sediment P sorption, likely due to the variable composition of Fe species present in these pools (Herndon et al., 2019; Senn et al., 2015). This result contrasts with previous notions that Fe:P indicates available sorption sites (Coelho et al., 2004; Jensen et al., 1992; Kronvang et al., 2009). Rather than the Fe:P ratio, our data suggests that the mass of BD-I Fe

alone was largely responsible for sediment P sorption (Fig. 7, Table S3). This result was consistent regardless of sediment texture, source of flow, or catchment geology. Notwithstanding the contribution of Al oxides to P sorption (Danen-Louwerse et al., 1993; Mendes et al., 2018), reactive Fe species (defined by varying methodologies) are strong predictors of P sorption in a variety of aquatic environments (Machesky et al., 2010; Zhang and Huang, 2007; Zhou et al., 2005). For example, Marton and Roberts (2014) and Herndon et al. (2019) found that the Bache-Williams index (BWI) was predicted by amorphous or

reactive Fe oxide contents in peat, tundra, and marsh soils; these observations are matched by our findings in benthic stream





sediments (Fig. 7). While these environments differ substantially, a similar theme is apparent regarding Fe-P relationships: where redox interfaces generate amorphous Fe oxides, there is a greater potential for P adsorption.

### 4.4 The ability for sediments to influence dissolved reactive phosphorus depends on reactivity and connectivity

The above discussion infers that reactive Fe oxides are efficient traps for P but in reality – due to the advective nature of the
hyporheic zone and the high spatiotemporal variation in biogeochemistry – the Fe oxides themselves are likely transient (van der Grift et al., 2014; Runkel et al., 1999; Smolders et al., 2017). Our study suggests that DRP paradoxically tends to increase with P sorption potential (Fig. 8), but only for the hill-fed streams. We suspect that, given the P sorption potential in these streams was driven by amorphous Fe oxides (Fig. 7), the dynamics of Fe cycling may be coupled with DRP through: 1) dynamic precipitation and dissolution of Fe oxides and their bound P (Rhoton et al., 2002; Runkel et al., 1999; Smolders et
al., 2017) and 2) the generation of Fe colloids at redox interfaces bearing P and elevating DRP (Baken et al., 2016a; Briggs et al., 2015; Gottselig et al., 2017; Ren and Packman, 2005). Both of these mechanisms are constrained for streams with little hyporheic exchange (e.g., the spring-fed streams in the present study; Boano et al., 2014). While we have indications of the former (Fig. 7 and Fig. 8) and latter (ferrihydrite SI correlation with $HPO_4^{2-}$ activity; Fig. S1) hypotheses, it is beyond the scope of our study to discuss these meaningfully.

Further research is needed to link the abiotic P exchange mechanisms we have discussed to the spatiotemporal DRP signal observed in the water column. While stream sediments are well-known to provide reaction sites for P, little has been done to link these reaction sites in streams to P cycling in a mechanistic manner, i.e., by connecting these zones of reactivity to hydrological transport (Boano et al., 2014). We suggest that future research on stream P cycling avoids mono-causal interpretations (Kalbitz et al., 2017), and incorporates the multiple mechanisms – both biotic and abiotic – where relevant.

## 5 Conclusions

Water-column and sediment biogeochemistry changes with catchment geology in ways critical to P cycling. Low to moderate alkalinity (i.e., <100 mg $L^{-1}$ as $CaCO_3$) streams had little potential to remove P via Ca-based (co-)precipitation mechanisms, making interactions with benthic sediments more important. We found sediment P sorption mechanisms captured within labile sediment P fractions ($H_2O$ and BD-I extractable P) were likley linked to reactive Fe pools that comprised minor proportions
of the total Fe content. The amount of this labile sediment P was predominantly driven by the catchment geology. However, the data also suggested that labile P co-varied with DRP where hyporheic exchange likely mediated the interaction between the water column and porewaters in hill-fed streams, whose sediments were more permeable than those in spring-fed streams. Both geology and hyporheic exchange are integral to the attenuation of P in streams and, hence, will need to be accounted for if baseflow DRP concentrations are to be predicted and managed.



## Data availability

The data reported here are available at Figshare (https://figshare.com/s/718226c7f1940d631755). The REC geodatabase is available through the New Zealand Ministry for the Environment (https://data.mfe.govt.nz/data/).

## Author contribution

Conceptualization: ZS and RM with support from LM. Funding acquisition: RM. Resources & supervision: RM and LC. Methodology: ZS, RM, and LC. Formal analysis: ZS with support from RM. Investigation, visualization & original draft: ZS. All authors contributed to the review and final draft.

## Competing interests

The authors declare that they have no conflict of interest.

## Acknowledgements

We thank Julie Clark at University of Otago for analyzing sediments for particle size distributions; Lincoln University technical staff for providing laboratory needs and some analyses, particularly Roger Cresswell, Lynne Clucas, Leanne Hassall, Vicky Zhang, and Shiv Pokhrel; Florencia De Lucca Agrelo for field and lab assistance; Marion Des Roseaux, Yuan Li, David Rex, and Tihana Vujinovic for occasional support; and the Our Land and Water National Science Challenge for funding the study (contract C10X1507 from the Ministry of Business, Innovation and Employment). This manuscript benefitted from review by Andrea Leptin.



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



**Table 1. Experimental procedure of the sequential sediment P fractionation, following the method of Jan et al. (2015); the targeted biogeochemical pools of P are given, but are not exact since fractionation methods are operationally defined. P and Fe analyses in the BD and NaOH fractions refer to total P and Fe. †Not applicable**


| Step or fraction | Solution | Extraction Time | Analyses | Primary biogeochemical pool |
|---|---|---|---|---|
| **1. H₂O** | D.I. water | 30 min | P | Labile, loosely-bound P |
| **2. BD-I** | Bicarbonate-dithionite | 5 + 5 min | P, Fe | Less-crystalline, surface-active Fe oxides |
| **3. BD-II** | Bicarbonate-dithionite | 2 h | P, Fe | Crystalline, poorly active Fe oxides |
| **Wash** | NaCl | 5 min | NA† | NA |
| **4. NaOH-I** | NaOH | 5 + 5 min | P, Fe | Active Al oxides, labile organic matter, clay minerals |
| **5. NaOH-II** | NaOH | 16 h | P, Fe | Crystalline Al oxides, refractory organic matter, clay minerals |
| **Wash** | NaCl | 5 min | NA | NA |
| **6. HCl** | HCl | 24 h | P | Primary minerals |





**Table 2. Summary of stream water chemistry grouped by River Environment Classification geology class; values are given as medians (means ± standard deviation); lowercase letter exponents represent group-wise comparisons between geology classes (confidence level = 0.95).**

| | Units | Geology Class | | |
|---|---|---|---|---|
| | | Alluvium (n=15) | Sedimentary (n=10) | Volcanic basic (n=6) |
| **pH** | S.U. | $7.29^a$ (7.26 ± 0.33) | $7.54^b$ (7.61 ± 0.22) | $7.62^b$ (7.62 ± 0.06) |
| **Conductivity** | µS cm$^{-1}$ | 142 (154 ± 51) | 78 (121 ± 79) | 155 (152 ± 18) |
| **Alkalinity** | mg L$^{-1}$ as CaCO$_3$ | 42.1 (44.3 ± 11.6) | 31.4 (42.9 ± 23.8) | 39.4 (38.2 ± 6.8) |
| **DRP** | µg L$^{-1}$ | $7.4^b$ (10.8 ± 7.6) | $5.0^a$ (5.6 ± 2.8) | $27.7^c$ (26.9 ± 11.3) |
| **NO$_3$-N** | mg L$^{-1}$ | $1.947^b$ (1.857 ± 1.22) | $0.111^a$ (0.626 ± 1.18) | $0.152^a$ (0.151 ± 0.080) |
| **SO$_4$** | mg L$^{-1}$ | $8.21^b$ (8.44 ± 2.77) | $4.89^b$ (9.38 ± 7.44) | $3.54^a$ (3.49 ± 0.55) |
| **Dissolved Fe** | mg L$^{-1}$ | $0.011^a$ (0.06 ± 0.139) | $0.011^a$ (0.012 ± 0.006) | $0.137^b$ (0.153 ± 0.069) |
| **Dissolved Ca** | mg L$^{-1}$ | $18.1^b$ (18.3 ± 5.52) | $11.9^b$ (17.09 ± 9.8) | $8.9^a$ (8.79 ± 1.33) |





**Table 3. Select physicochemical properties of the stream benthic sediments grouped by River Environment Classification geology class; D$_{50}$ is the median particle size of the fine sediments (<2 mm); values are given as medians (means ± standard deviation); lower case letter exponents represent group-wise comparisons between geology classes (confidence level = 0.95).**

| | Units | Geology Class | | |
|---|---|---|---|---|
| | | Alluvium (*n*=15) | Sedimentary (*n*=10) | Volcanic basic (*n*=6) |
| **pH** | S.U. | 6.93[a] (6.91 ± 0.24) | 7.12[ab] (7.26 ± 0.46) | 7.26[b] (7.31 ± 0.26) |
| **Total C** | g kg$^{-1}$ | 3.68[b] (13.2 ± 17.9) | 1.06[a] (1.3 ± 0.58) | 8.53[b] (10.8 ± 5.9) |
| **Total N** | g kg$^{-1}$ | 0.41[b] (1.31 ± 1.69) | 0.20[a] (0.23 ± 0.12) | 0.65[b] (0.83 ± 0.43) |
| **Total Al** | g kg$^{-1}$ | 20.15[a] (21.93 ± 6.04) | 18.44[a] (20.42 ± 4.88) | 35.39[b] (37.2 ± 8.35) |
| **Total Ca** | g kg$^{-1}$ | 5.77[a] (6.25 ± 2.34) | 5.97[ab] (6.35 ± 2.21) | 9.59[b] (9.46 ± 2.14) |
| **Total Fe** | g kg$^{-1}$ | 22.14[a] (21.95 ± 4.05) | 23.31[a] (23.44 ± 3.36) | 48.45[b] (50.43 ± 5.93) |
| **Total Mn** | mg kg$^{-1}$ | 379[a] (455 ± 275) | 372[a] (368 ± 89.2) | 725[b] (713 ± 160) |
| **Total P** | mg kg$^{-1}$ | 548[a] (597 ± 219) | 476[a] (462 ± 121) | 2130[b] (2220 ± 525) |
| **Clay** | % volume | 0.011[b] (0.99 ± 1.6) | 0[a] (0.001 ± 0.004) | 0.3[b] (0.5 ± 0.7) |
| **Silt** | % volume | 5.25[b] (19.8 ± 25.9) | 0.049[a] (0.98 ± 1.54) | 14.7[b] (16.8 ± 9.96) |
| **Sand** | % volume | 94.7[a] (79.2 ± 27.4) | 99.9[b] (99.0 ± 1.55) | 85.0[a] (82.7 ± 10.7) |
| **D$_{50}$** | µm | 415[a] (356 ± 255) | 701[b] (680 ± 232) | 544[ab] (491 ± 238) |





**Table 4. Stream sediment P fractions, Fe fractions, molar Fe:P ratios (including total Fe to total P), and sorption metrics. Values are given as medians (means ± standard deviation). The lowercase letter exponents represent group-wise comparisons between geology classes (confidence level = 0.95); for explanation of the sorption metrics, see Sect. 2.**

| | Units | Geology Class | | |
|---|---|---|---|---|
| | | Alluvium (n=15) | Sedimentary (n=10) | Volcanic basic (n=6) |
| **Sediment P fractions** | | | | |
| **H₂O** | mg P kg⁻¹ | 1.39[b] (1.82 ± 1.41) | 0.417[a] (0.436 ± 0.081) | 2.48[b] (3.84 ± 3.76) |
| **BD-I** | | 27.8[b] (52.1 ± 51.9) | 4.14[a] (3.62 ± 1.87) | 31.8[b] (40.4 ± 25.5) |
| **BD-II** | | 14.8[b] (32.2 ± 32.6) | 3.53[a] (2.98 ± 1.405) | 38.4[b] (37.2 ± 4.52) |
| **NaOH-I** | | 39.4[b] (61.2 ± 60.3) | 25.2[a] (23.8 ± 7.11) | 292[c] (342 ± 185) |
| **NaOH-II** | | 20.2[ab] (30.6 ± 29.1) | 10.5[a] (10.4 ± 4.79) | 97.2[b] (176 ± 177) |
| **HCl** | | 271[a] (337 ± 163) | 414[ab] (452 ± 161) | 1400[b] (1370 ± 572) |
| **Sediment Fe fractions** | | | | |
| **BD-I** | mg Fe kg⁻¹ | 529[b] (1360 ± 1500) | 252[a] (254 ± 66.1) | 2480[c] (2560 ± 623) |
| **BD-II** | | 664[a] (915 ± 930) | 433[a] (446 ± 190) | 5990[b] (5150 ± 1670) |
| **NaOH-I** | | 50.4[b] (92.98 ± 102) | 21.7[a] (22.6 ± 8.18) | 107[c] (118 ± 41.6) |
| **NaOH-II** | | 447[b] (122 ± 182) | 35.9[a] (36.2 ± 11.1) | 99.6[b] (131 ± 94.6) |
| **Fe:P (molar)** | | | | |
| **BD-I** | mol Fe mol P⁻¹ | 15.5[a] (18.3 ± 9.85) | 36.4[b] (65.01 ± 66.9) | 39.1[b] (40.6 ± 14.3) |
| **BD-II** | | 19.9[a] (31.6 ± 25.05) | 80.7[b] (99.2 ± 61.01) | 85.8[b] (76.6 ± 21.6) |
| **NaOH-I** | | 0.71[a] (1.15 ± 1.15) | 0.52[b] (0.59 ± 0.34) | 0.23[b] (0.25 ± 0.15) |
| **NaOH-II** | | 2.11 (2.69 ± 2.40) | 1.80 (2.47 ± 1.63) | 0.38 (3.12 ± 6.25) |
| **Total** | | 22.3[b] (22.2 ± 5.73) | 28.4[c] (29.1 ± 4.25) | 13.3[a] (13.1 ± 2.64) |
| **Sorption metrics** | | | | |
| **ASC** | % P retention | 14.3[b] (33.6 ± 26.9) | 6.98[a] (10.2 ± 5.58) | 49.4[b] (49.4 ± 11.1) |
| **BWI** | $\dfrac{\text{mg P kg}^{-1}}{\log_{10}(\mu\text{g P L}^{-1})}$ | 12.7[b] (51.95 ± 54.7) | 6.84[a] (8.8 ± 5.57) | 53.5[b] (53.4 ± 12.3) |







**Figure 1. Study streams, and their catchments, in Canterbury, New Zealand (see inset). The geology classification used by the (New Zealand) River Environment Classification scheme is shown, with 'Miscellaneous' (loess, peat), 'Other' (river beds, ice cover, lakes, and urban centers), and Plutonic geology classes excluded for clarity. Sampling locations are shown, with circles indicating 'Hill-fed' streams (i.e., no major groundwater inputs) and triangles indicating 'spring-fed' streams.**





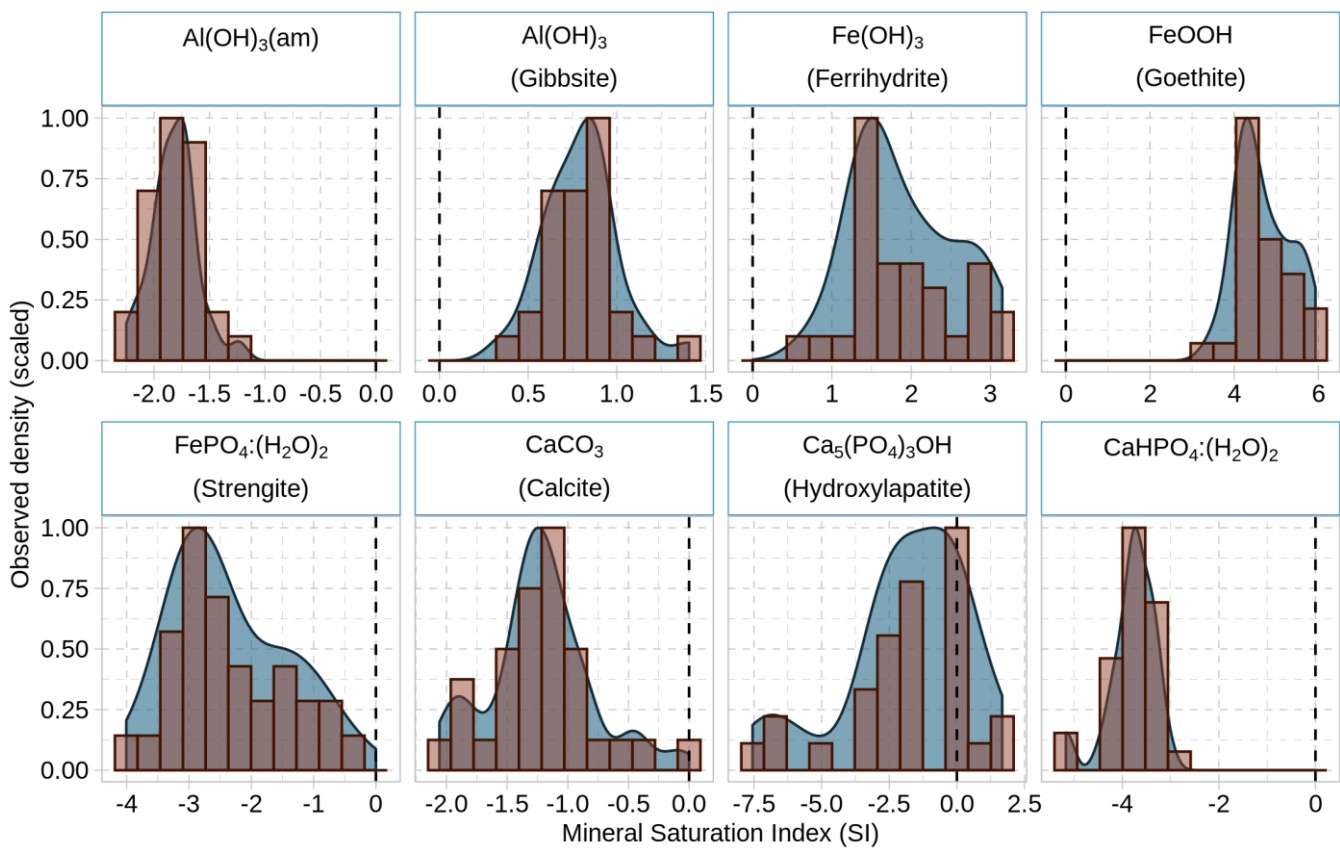


**Figure 2. Distributions of select mineral saturation indices for the stream samples ($n$=31), where positive (negative) saturation index (SI) indicates the thermodynamic potential for the mineral to precipitate (dissolve). The frequency distributions are displayed as normalized densities. SI=0 is indicated on each sub-plot with a dashed vertical line. The solid phase chemical formulae are provided as given in the MINTEQ.v4 database.**



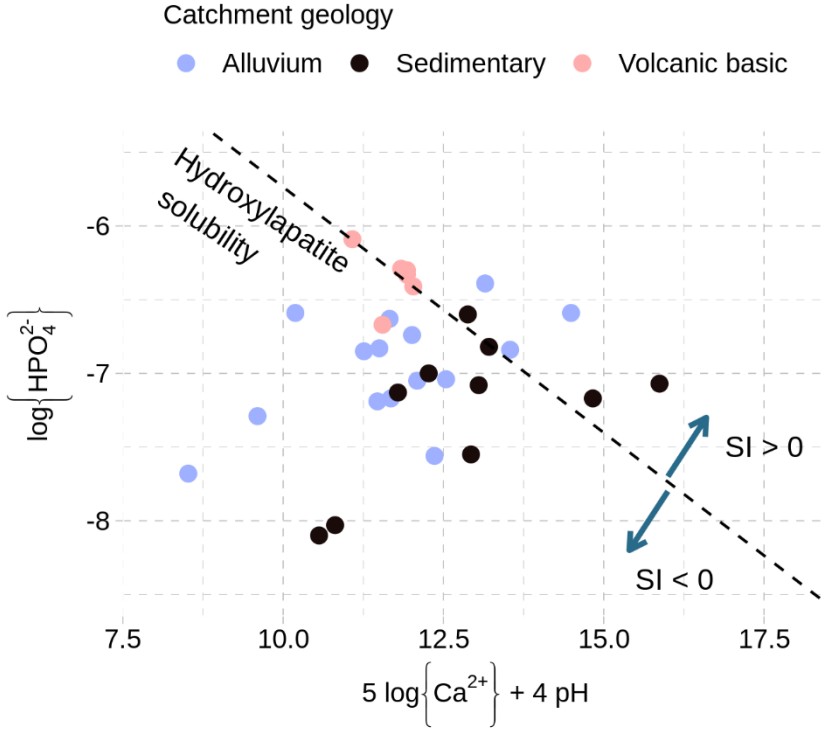

**Figure 3. Log-activity of HPO₄²⁻ plotted against a function of log-activities of Ca²⁺ and H⁺ for the stream samples (*n*=31). The dashed line is a reference solubility line for hydroxylapatite, where points below (above) this line are likely sub-saturated (super-saturated) with respect to hydroxylapatite.**



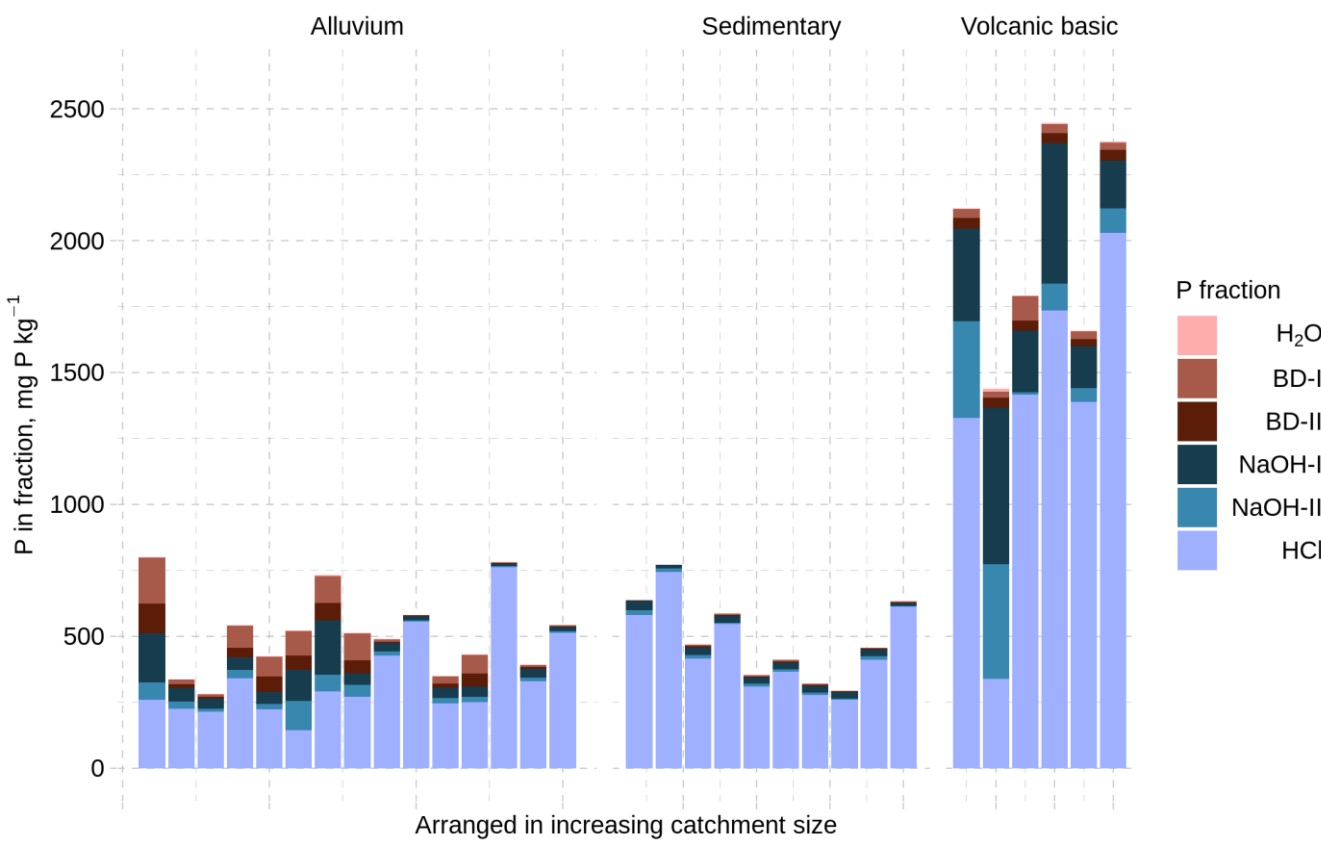

**Figure 4. Benthic sediment (<2 mm) phosphorus content fractionated according to decreasing chemical lability; bars are arranged in increasing order of catchment size within the River Environment Classification geology class.**








**Figure 5. Stream DRP concentration as a function of sediment H$_2$O-P and sediment labile P (H$_2$O-P plus BD-I P) concentrations from the sequential P fractionations. Spearman correlation tests are shown for all data (*n*=31) as well as for only hill-fed streams (*n*=23).**



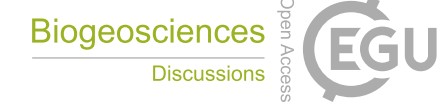

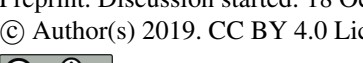


**Figure 6. Sediment Fe concentration as measured in the P fractions (potentially reactive Fe arranged in decreasing lability; top row) or as the total Fe concentration (bottom row); note the differences in scale. Each bar (site) is arranged in increasing catchment size within the River Environment Classification geology class.**







**Figure 7. Sediment P sorption potential as measured by anion storage capacity (ASC; %) as a function of sediment Fe in the bicarbonate-dithionite (BD) extractable pools and total sediment Fe. The optimal robust linear models for ASC with each Fe fraction, as discussed in text, are shown; note that while BD-I Fe alone predicts ASC, the models for BD-II Fe and total Fe include catchment geology as a covariate.**



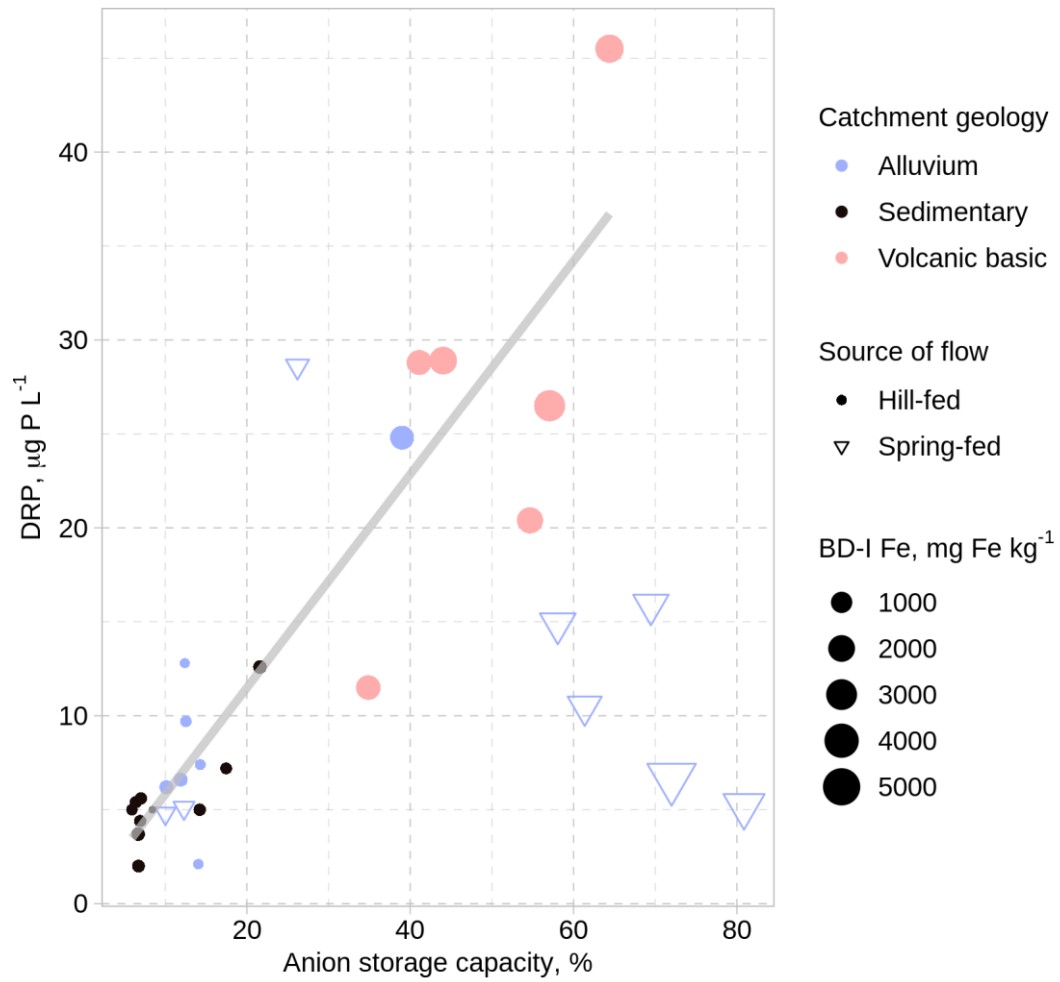

**Figure 8. In-stream DRP as a function of sediment P sorption metrics (anion storage capacity (ASC; %) and Bache-Williams Index (BWI; refer to methods for unit interpretation)). The robust linear model discussed in text is shown.**