# Peer review of "Phosphorus attenuation in streams by water-column geochemistry and benthic sediment reactive iron"

_Biogeosciences, 2019_

## Referee Comment (RC1) · Anonymous Referee #1 · 26 Nov 2019

General Comments: In the manuscript entitled "Phosphorus attenuation in streams by water-column geochemistry and benthic sediment reactive iron," researchers assessed how excess phosphorus can be retained in the sediments through geochemical processes. Research on abiotic phosphorus retention in river networks draining watersheds with varying land cover and geology is lacking. This information would be useful to managers all over the world who are trying to find ways to improve nutrient removal in disturbed watersheds. The manuscript is well written, and few changes are necessary before it is ready to be published. Specific Comments: Line 76: Did you have any hypotheses about how the geochemical processes may differ between the three different stream geology types? Table S2. In the results section, you refer to this

table twice. It was difficult to understand in lines 245-247 why you picked the Geology, H2O-P model as the best model. For clarity, I would make a note indicating that this is the best model when ASC is excluded. In many instances in the results, you restated results that are displayed in tables or figures. I have indicated these instances in the technical corrections. Technical Corrections: Line 141: What does "d.w." mean? Line 152: Change "RP" to "Reactive P". Line 171: Remove colon Lines 198-199: Delete concentrations and instead reference Table 2. Lines 210-211: Delete "log-activity of . . . . . ..in Fig. 3" and just reference Fig. 3. Line 218: Delete first sentence of section 3.2. Line 218: Change to "The benthic sediments were largely neutral (mean pH of 7.10; Table 3)." Lines 228-229: Delete all the mean values and just reference Table 4. Lines 241-242: Change to "The least available sediment P pool analyzed, HCl-P was highest in volcanic basin and lowest in alluvium geologies." Line 245: Change sentence to "Using geology and sediment P pools. . . .." Line 250: Delete means and just reference Table 3. Lines 268-269: Delete all the values and just reference Table 4. Lines 271-272: Delete the first sentence in the section. Move "ASC(a measure of P retention at low pH) and BWI (P retention at neutral pH and controlled Ca concentration)" to the methods section 2.4. Lines 274-275: Delete values and just reference Table 4. Line 404: "likely" is misspelled. Figure 3: Indicate that SI = saturation index.

---

## Referee Comment (RC2) · Anonymous Referee #2 · 9 Dec 2019

With apologies for my late review. With interest I have read this manuscript, in which the authors explore P retention/precipitation mechanisms in New Zealand streams with contrasting underling geology (alluvium, bedrock, sediment). The authors combine geochemical modelling based on analysis of dissolved chemistry of stream waters with chemical sediment analysis (Fe, P fractionation) to infer the likelihood of mineral precipitation and to determine P species in the sediments. The authors find that sorption onto Fe minerals (presumably Fe (oxyhydr)oxides) play an important role in P retention in sediments that consist mostly of recalcitrant Fe and P phases. The paper is generally well-written.

[Figure]

I do find it difficult to fully appreciate the storyline in the paper. The introduction goes into some detail regarding eutrophication and P attenuation and the potential role of sediments. Then, quite suddenly it is mentioned that "therefore" (why?) the New Zealand streams in the Canterbury region have been investigated, because they "cover[] a variety of geologies, land use, and stream characteristics". Because of the heavy enphasis on eutrophication and attenuation in the paper, it would be good if these topic are touched upon in relation to the research area. At the moment, the two seem disconnected.

My second main points regards the balance between data and modelling. I like geochemical modelling as much as the next biogeochemist, and I think it is a very useful tool to explore the likelihood of certain precipitation mechanisms, and the validity of experimental findings. Here, I wonder about a few aspects of the approach. The authors use geochemical modelling with the stream data as input to investigate potential precipitation reactions in the stream bed material, with special emphasis on Fe(-P) and Ca(-P). The results and interpretation for Fe are, as the authors themselves mention, unreliable and probably related to measuring colloidal Fe-P phases that pass the 0.45 um filters (L327-335). Then, the authors also use stream chemistry to assess the likelihood of Ca-P precipitation. I wonder whether the overlying water is representative for the chemistry of the interstitial water, where mineral formation would occur? In many aquatic systems, there are large chemical differences between interstitial water and overlying water, resulting in large differences in saturation state with respect to relevant minerals. Given the emphasis on the role of hyporheic exchange, I wondered how this would work here. Is this also of relevance for the investigated stream sediments, or do the authors have good reasons to assume stream water is representative for all water in the system?

The authors use an extraction protocol (NaOH, BD, HCl) that is not very suitable to investigate authigenic Ca-P phases in the sediment. The HCl extraction removes large amounts of P that was supposedly associated with recalcitrant, geological apatite. I find

the investigation of the possibility of Ca-P formation in the stream beds rather poorly developed with geochemical modelling of stream water and not interstitial water, and chemical extractions insuitable to explore authigenic Ca-P formation. The uncertainty around apatite exists throghout the paper. In the abstract, the authors mention that apatite does not occur in those streams. Then, in L304 it is mentioned that Ca-P formation is a best a secondary mechanism for P removal. Those are quite contrasting statements, and illustrate for me the unclear and unsatisfactory way in which Ca-P precipitation is deal with in the paper.

The main finding of the paper then appears to be that Fe (oxyhydr)oxides play an important role in P retention in these streams, and that reactive Fe abundance is a function of geology of the stream bed. In L362 and further, the authors then introduce literature from the field of Fe(II) oxidation and the resulting formation of a range of P-bearing Fe species (citation of the work of Senn et al. 2015), which seems quite out of context in this story as Fe redox cycling seems not to be a part of it. If it is, then the hyporeic zone is a very different geochemical zone altogether and using the stream water to explore mineral formaion makes little sense... To me, this is all a bit symptomatic of how bits and pieces of geochemistry are used in hypothetical and sometimes inappropriate ways, while there is relativey little robust data (solid-phase Fe and P fractionation, geochemical modelling of the stream water) to really pinpoint the processes governing P cycling and retention in the stream beds. The fact that poorly crystalline Fe (oxyhydr)oxides are important in P retention is very well-established, and therefore I have my doubts whether the findings in this contribution provide enough novel insight. The authors mention that this is the first time that the extraction potocol of Jan et al. (2015) is used, but that novelty is of limited relevance to me, especially as I do not think much understanding on P burial in the stream beds is gleaned from it.

Lastly, nothing is really done with the findings. As mentioned, in the introduction it does not become clear what relevance the understanding of P retention mechanisms in the investigated streams has, and in the discussion and conclusion it does

not become apparent what the implications of the findings are. The findings fit with well-established concepts of Fe-controlled P mobility, but there is no link to the situation/future/management of the investigated streams.

Overall, this reviewer is left wondering whether the identification of a well-established P retention mechanism in New Zealand streams, while the finer details of P burial are obscured because of limited data and dependence on uncertain geochemical modelling, warrants publication in BG. Especially as neither the context for studying this area, nor the implications of the findings, are very well-developed.

---

## Author Comment (AC1) · 16 Jan 2020

Thank you to both referees for their thoughtful reviews. We address comments from both reviews here in turn.

Following the original comments in *italic text*, we give our responses in plain blue text and any resulting revisions to the manuscript in plain red text.

**Anonymous referee #1:**

*General Comments: In the manuscript entitled "Phosphorus attenuation in streams by water-column geochemistry and benthic sediment reactive iron," researchers assessed how excess phosphorus can be retained in the sediments through geochemical processes. Research on abiotic phosphorus retention in river networks draining watersheds with varying land cover and geology is lacking. This information would be useful to managers all over the world who are trying to find ways to improve*

*nutrient removal in disturbed watersheds. The manuscript is well written, and few changes are necessary before it is ready to be published.*

Thank you for the thorough review, particularly for helping with making the paper more concise. We took the opportunity to try to accordingly remove unnecessary data references and make the text, in general, more readable. See also our specific changes below.

*Specific Comments: Line 76: Did you have any hypotheses about how the geochemical processes may differ between the three different stream geology types?*

Thank you for pointing this out. We sampled varying geologies because of our tacit hypothesis that both sediment and water chemistry would vary between geologies (e.g., chalk catchments would have potential calcite co-precipitation with phosphate and geologies rich in cations (especially Fe) would provide highly sorptive sediments), and so provide an interesting gradient for exploring potential abiotic P attenuation mechanisms. We have taken the opportunity to refine the hypotheses and objectives of the work, making our motivation more explicit (starting on line 73):

We hypothesized that the primary abiotic mechanisms responsible for P attenuation (and therefore related to DRP concentrations) were Ca-based mineral equilibria in the water column and sorption with benthic sediments. Under this hypothesis, we expected that the prominence of either mechanism would be tied to geology: Ca-P (co-)precipitation would be more likely for streams draining calcareous geologies and sediments would be more sorptive for P when originating from geologies rich in Fe and Al minerals. Further, we hypothesized that amorphous, reactive Fe was a primary controller of sediment P sorption, rather than refractory or total Fe pools. Specifically, our first objective was to identify whether Ca-P (co-)precipitants were favorable in streams draining calcareous geologies (here, sedimentary geologies) and, hence, a potential mechanism for P attenuation. Our second objective was to identify predominant pools of P in the sediments in relation to their potential lability (governed by the biogeochemistry represented, e.g., sensitivity to redox) and how they vary between catchments. Our third objective was to model sediment P sorption as a function of Fe fractions to test whether this mechanism for P attenuation varied with the reactivity of the Fe.

*Table S2. In the results section, you refer to this table twice.*

We removed the second reference in line 287.

*It was difficult to understand in lines 245-247 why you picked the Geology, H2O-P model as the best model. For clarity, I would make a note indicating that this is the best model when ASC is excluded.*

We see now that this section was opaque. We've revised the text as suggested and clarified how the best-fit model (excluding ASC for the moment) was selected:

Using geology and sediment P pools as the primary predictors for DRP in hill-fed streams (excluding P sorption potential, see below; Table S2), the best-fit linear model employed geology and $H_2O$-P (RMSE = 3.93 µg P $L^{-1}$), with a slope (95% C.I.) of 2.62 (1.58 to 3.66) µg P $L^{-1}$ per mg P $kg^{-1}$ for $H_2O$-P. Competing models either had much greater AIC (simpler models with either geology or $H_2O$-P only) or similar AIC but more model degrees of freedom (model with geology, $H_2O$-P, and BD-I P).

*In many instances in the results, you restated results that are displayed in tables or figures. I have indicated these instances in the technical corrections.*

We have attempted to thin these down to only essential references, in addition to specific corrections later.

*Technical Corrections: Line 141: What does "d.w." mean?*

We have revised the text to "We used 0.5 g (dry weight) sediment …"

*Line 152: Change "RP" to "Reactive P".*
*Line 171: Remove colon*
*Lines 198-199: Delete concentrations and instead reference Table 2.*
*Lines 210-211: Delete "log-activity of….in Fig. 3" and just reference Fig. 3.*
*Line 218: Delete first sentence of section 3.2.*
*Line 218: Change to "The benthic sediments were largely neutral (mean pH of 7.10; Table 3)."*
*Lines 228-229: Delete all the mean values and just reference Table 4.*

Thank you for these improvements. Text has been revised as suggested.

*Lines 241-242: Change to "The least available sediment P pool analyzed, HCl-P was highest in volcanic basin and lowest in alluvium geologies."*

That sentence has been revised to "The least available sediment P pool analyzed, HCl-P, was greatest in volcanic basic sediments and similar between sedimentary and alluvium sediments."

*Line 245: Change sentence to "Using geology and sediment P pools…"*
*Line 250: Delete means and just reference Table 3.*
*Lines 268-269: Delete all the values and just reference Table 4.*

Thank you, text has been revised as suggested.

*Lines 271-272: Delete the first sentence in the section. Move "ASC (a measure of P retention at low pH) and BWI (P retention at neutral pH and controlled Ca concentration)" to the methods section 2.4.*
*Lines 274-275: Delete values and just reference Table 4.*

We have moved that text to the methods as suggested so that the first paragraph in section 2.4 ends with "…the supernatants were analyzed for DRP via the molybdenum-blue method (Murphy and Riley, 1962). Hence, ASC is a measure of P retention at low pH while BWI is a measure of P retention at neutral pH and controlled Ca concentration."

Likewise, the first paragraph of section 3.4 has been revised to:

Although ASC and BWI differ in how P sorption potential is determined, a similar pattern was apparent in both variables for the three sampled geologies. Given the widespread use of ASC in soil classification (and management) in New Zealand (Saunders 1965), we focus on ASC for brevity. The sedimentary samples had lower ASC than either alluvium or volcanic
basic sediments (Table 4). There was no clear relationship between ASC and sediment Fe:P ratios (Fig. S6), while ASC was correlated with BD-I ($\rho = 0.77$, p <0.001), BD-II ($\rho = 0.65$, p <0.001), and total Fe ($\rho = 0.55$, p =0.002) pools (Fig. 7).

*Line 404: "likely" is misspelled.*

Thanks for the catch! Revised.

*Figure 3: Indicate that SI = saturation index.*

The second sentence in the caption for Figure 3 has been revised to:
The dashed line is a reference solubility line for hydroxylapatite, where points below (above) this line are likely sub-saturated (super-saturated) with respect to hydroxylapatite, as indicated by the saturation index (SI).

**Anonymous referee #2:**

*With apologies for my late review. With interest I have read this manuscript, in which the authors explore P retention/precipitation mechanisms in New Zealand streams with contrasting underling geology (alluvium, bedrock, sediment). The authors combine geochemical modelling based on analysis of dissolved chemistry of stream waters with chemical sediment analysis (Fe, P fractionation) to infer the likelihood of mineral precipitation and to determine P species in the sediments. The authors find that sorption onto Fe minerals (presumably Fe (oxyhydr)oxides) play an important role in*
*P retention in sediments that consist mostly of recalcitrant Fe and P phases. The paper is generally well-written.*

*I do find it difficult to fully appreciate the storyline in the paper.*

We thank the referee for their critical comments concerning the coherence of the paper. Indeed, we wish our 'story' to be apparent to the reader and are glad for this opportunity to refine it. In particular, with this review, we have sharpened the discussion concerning mineral equilibria and stream P attenuation (see later) and consider the paper better for it.

In general, we believe we can address the referee's main concerns (more specific below) by: 1) making some concepts/connections more explicit, e.g., in the introduction and when discussing the potential role of hyporheic exchange, 2) clarifying the relevance of P fractionation, what our choice of fractionation scheme does get right (all fractionation schemes, in a sense, fall short), 3) acknowledging that alternative P fractionations would complement our findings on Ca-P through mineral equilibria, 4) make redox a more recurring theme throughout the paper so it does not surprise the reader later in the discussion, and 5) make explicit what our findings imply for P pollution mitigation in streams.

*The introduction goes into some detail regarding eutrophication and P attenuation and the potential role of sediments. Then, quite suddenly it is mentioned that "therefore" (why?) the New Zealand streams in the Canterbury region have been investigated, because they "cover[] a variety of geologies, land use, and stream characteristics". Because of the heavy enphasis on eutrophication and attenuation in the paper, it would be good if these topic are touched upon in relation to the research area. At the moment, the two seem disconnected.*

We have re-worked the last few paragraphs of the introduction so that our logical flow is more sensible. Additionally, in response to Referee #1's comments above, we have clarified the hypotheses/objectives towards the end of the introduction. Regarding how eutrophication fits into the study area context, we briefly mention the case for Canterbury, NZ and supply a reference for more background (McDowell et al., 2013; cited already in the manuscript). Hopefully, the inter-relation between eutrophication and P attenuation is clearer and provides our motivation for the study.

Additionally, relating to concerns given below, we more strongly work in the theme of redox and amorphous Fe oxides in the streams as strong sorption sites for P. We think this is still an emerging concept for streams. Much stream research uses inadequate measures for this topic (often, sediment total Fe gets used in regression-type analyses to infer controls on P). Our work here shows that total Fe is a poor predictor of sorption when compared to amorphous Fe oxides. We know of little other work in streams that focuses on these Fe fractions, much less show the strong relationship between sediment P sorption and this specific Fe fraction which we observed. We attempted to update the introduction to better prepare the reader for that result and its significance.

The remainder of the introduction starting with line 57 has been revised to:

Benthic stream sediments can have large potential to adsorb P and thus attenuate P inputs (Froelich, 1988; Haggard and Sharpley, 2007; McDowell, 2015), especially for baseflow conditions where water is given time to contact sediments in the hyporheic zone (Harvey, 2016). This sediment sorption can be readily examined via intensity of adsorption and the quantity of P already complexed with the sediment. Sorption intensity measurements often correlate negatively with in-stream DRP (McDaniel et al., 2009; McDowell, 2015; Weigelhofer et al., 2018) and positively with stream P uptake metrics (Demars, 2008; Haggard et al., 2005; Jarvie et al., 2005). Concordantly, sediments in streams with high P loading (i.e., high sustained DRP concentrations) tend to have diminished sorption ability and greater stores of P (Jarvie et al., 2012; McDowell, 2015), particularly in the more labile and redox-sensitive pools (Lewandowski and Nützmann, 2010; Weigelhofer, 2017). However, these redox-sensitive pools (i.e., iron (Fe) oxy(hydr)oxides, henceforth Fe oxides) are poorly studied for in-stream P attenuation.

Fe oxides are one pool among the reactive surfaces responsible for P sorption; notably, clay minerals and other metal (i.e., aluminum) oxides are strong reaction sites (Gérard, 2016; Golterman, 2004; Parfitt, 1979). However, Fe oxides may compose the more prominent reaction sites in many non-calcareous streams (Dupas et al., 2018b; van der Grift et al., 2014; Lewandowski and Nützmann, 2010). Particularly, amorphous, surface-reactive Fe oxides – e.g., poorly-crystalline goethite, lepidocrocite, and ferrihydrite (Jan et al., 2015; Stumm and Sulzberger, 1992) – have the greatest affinity for P (Goldberg and Sposito, 1984; Lijklema, 1980) but are a variable and potentially minor fraction of the total sediment Fe concentration (Hyacinthe et al., 2006; Jan et al., 2013; Parsons et al., 2017). Further, fresh (amorphous) Fe oxides may form along the redox gradient in the hyporheic zone but this is likely stream dependent, e.g., hydraulic and physicochemical characteristics may interact with geological Fe supply, thus varying the mass of amorphous Fe oxides among streams (Boano et al., 2014; Smolders et al., 2017). Much contemporary stream sediment research overlooks this and relies on total Fe or roughly defined Fe fractions to discuss Fe-P interactions, thus hampering our understanding of how Fe oxides affect DRP concentrations (Hoffman et al., 2009; Kreiling et al., 2019; Rawlins, 2011; Tye et al., 2016). In other environments, amorphous Fe oxides are increasingly recognized as dominant P sorption sites (Marton and Roberts, 2014; Parsons et al., 2017; Zhang and Huang, 2007). We hypothesize this to apply to stream benthic sediments as well.

In the present study, we examined P attenuation mechanisms in streams at baseflow via Ca-P geochemistry in the water column, stores of sediment P and redox-sensitive Fe, and P sorption capacities of sediments. Given that these processes are likely all tied to catchment geology and P inputs, we sampled waters and benthic sediments of streams in the Canterbury region, New Zealand, where streams cover a variety of geologies, land use, and stream characteristics and differ in typical baseflow DRP concentrations (McDowell et al., 2013). We hypothesized that the primary mechanisms responsible for P attenuation (and therefore related to DRP concentrations) were Ca-based mineral equilibria in the water column and sorption with benthic sediments. Under this hypothesis, we expected that the prominence of either mechanism would be tied to geology: Ca-P (co-)precipitation would be more likely for streams draining calcareous geologies and sediments would be more sorptive for P when originating from geologies rich in Fe and Al minerals. Further, we hypothesized that amorphous, reactive Fe was a primary controller of sediment P sorption, rather than refractory or total Fe pools. Specifically, our first objective was to identify whether Ca-P (co-)precipitants were favorable in streams draining calcareous geologies (here, sedimentary geologies) and, hence, a potential mechanism for P attenuation. Our second objective was to identify predominant pools of P in the sediments in relation to their potential lability (governed by the biogeochemistry represented, e.g., sensitivity to redox) and how they vary between catchments. Our third objective was to model sediment P sorption as a function of Fe fractions to test whether this mechanism for P attenuation varied with the reactivity of the Fe.

Additional references cited and added to manuscript:

Hyacinthe, C., Bonneville, S. and Van Cappellen, P.: Reactive iron(III) in sediments: Chemical versus microbial extractions, Geochim. Cosmochim. Acta, 70(16), 4166–4180, doi:10.1016/j.gca.2006.05.018, 2006.

Kreiling, R. M., Thoms, M. C., Bartsch, L. A., Richardson, W. B. and Christensen, V. G.: Complex Response of Sediment Phosphorus to Land Use and Management Within a River Network, J. Geophys. Res. Biogeosci., 0(0), doi:10.1029/2019JG005171, 2019.

Rawlins, B. G.: Controls on the phosphorus content of fine stream bed sediments in agricultural headwater catchments at the landscape-scale, Agriculture, Ecosystems & Environment, 144(1), 352–363, doi:10.1016/j.agee.2011.10.002, 2011.

Tye, A. M., Rawlins, B. G., Rushton, J. C. and Price, R.: Understanding the controls on sediment-P interactions and
dynamics along a non-tidal river system in a rural-urban catchment: The River Nene, Appl. Geochem., 66, 219–233, doi:10.1016/j.apgeochem.2015.12.014, 2016.

Weigelhofer, G.: The potential of agricultural headwater streams to retain soluble reactive phosphorus, Hydrobiol., 793(1), 149–160, doi:10.1007/s10750-016-2789-4, 2017.

*My second main points regards the balance between data and modelling. I like geochemical modelling as much as the next biogeochemist, and I think it is a very useful tool to explore the likelihood of certain precipitation mechanisms, and the validity of experimental findings. Here, I wonder about a few aspects of the approach. The authors use geochemical modelling with the stream data as input to investigate potential precipitation reactions in the stream bed material, with*
*special emphasis on Fe(-P) and Ca(-P).*

We respond to the specifics of this second point in-line (below).

*The results and interpretation for Fe are, as the authors themselves mention, unreliable and probably related to measuring*
*colloidal Fe-P phases that pass the 0.45 um filters (L327-335).*

As mentioned in the text, this was a shortcoming and we consider it a lesson learned. (We couldn't conceive of everything!) Fe colloids in streams – and especially their potential to carry P as a vector – is an emerging research topic within recent years. We think this shortcoming in our dataset actually highlights the need to study Fe colloids in future work on streams as
mentioned in section 4.4; we just don't dwell on it for the sake of space.

That said, we did not expect Fe-P mineral phases to be significant in these waters (that is, the formation of those minerals in the typical stream chemistry, discounting what goes on along redox gradients in the hyporheic zone). Some Fe-P minerals can be important in other aquatic environments (e.g., vivianite in lake or very deep river sediments; Tye et al. 2016). We
make no attempt to study the deeper porewaters in our streams (see below for further discussion) where other P sequestration phenomena may occur but certainly operate on a much longer timescale than P attenuation under baseflow.

*Then, the authors also use stream chemistry to assess the likelihood of Ca-P precipitation. I wonder whether the overlying water is representative for the chemistry of the interstitial water, where mineral formation would occur? In many aquatic*
*systems, there are large chemical differences between interstitial water and overlying water, resulting in large differences in saturation state with respect to relevant minerals. Given the emphasis on the role of hyporheic exchange, I wondered how this would work here. Is this also of relevance for the investigated stream sediments, or do the authors have good reasons to assume stream water is representative for all water in the system?*

We thank the reviewer for these questions. We think the matter is simply resolved and the text can be sharpened to address this concern in a clear manner. On working through a sufficient response here, we have focused our own conception of the relevant mechanism. We are grateful and will update the text to better explain the importance and relation of mineral equilibria in the water column to P attenuation.

First and foremost, mineral formation need not only occur in interstitial waters. For example, calcite deposition on macrophytes or benthic surfaces (biofilms) in streams is a well-known phenomenon (e.g., see the Golterman, 2004 reference in the text). A nice example specific to this point is found in Tobias and Böhlke (2011; now added to text), who focused on inorganic C cycling in streams, and found that carbonate mineral deposition during the day could be found in the first 5 mm of benthic sediment (i.e., the top of the benthic substrate). These authors and much other contemporary research on stream mineral equilibria mechanisms focus on – and sample – the water-column. Our sampling was influenced by this research and we have updated the text accordingly.

We agree with the reviewer on large chemical differences between various zones of water. However, since our study scope is on P attenuation observed on the minutes to hours timescales relevant to streams (e.g., as seen in nutrient spiraling studies), we do not concern ourselves with the deeper hyporheic zone or groundwater where residence times of water are much longer (not to say those zones are not important). There, chemical differences with respect to the water-column are the greatest (redox, pH, etc.). In the shallow hyporheic zone (e.g., first few cm), residence times are more within the scale of minutes to hours (Boano et al. 2014; cited in text). The chemistry gradients there are not so great as to prevent deposition of precipitants initialized in the water-column/armor layer.

Second, even if the potential precipitants would require the shallow sediment porewaters as space to deposit, the primary change in water chemistry within the hyporheic zone that would have a significant bearing on mineral equilibria (precipitation) is pH. We expect pH to decrease with depth in the hyporheic zone due to respiration. However, our sediment pH values (a rough proxy for pH in these porewaters; Table 3) are not that dissimilar from stream pH: mean pH in streams (sediments) were 7.26 (6.91), 7.54 (7.26), and 7.62 (7.31) for alluvium, sedimentary, and volcanic basic geologies, respectively. Similar to other stream studies, we suspect that precipitation would primarily occur at the most surficial benthic layer (e.g., within the armor layer in the gravel bedded streams) and that the pH from shallow benthic sediments would not necessarily override this.

We have updated the text to reflect this research and sharpen our conception of how mineral equilibria relates to P attenuation. In the paragraph beginning with L48, we have revised the text to:

Calcium-phosphate mineral precipitation and $CaCO_3$ co-precipitation may remove DRP from the water column given sufficient Ca, pH, and $p$CO2 (Golterman, 2004; House, 2003; Stumm and Morgan, 1996). Like carbonates and other minerals formed when the water-column is super-saturated in respect to those phases, these minerals could remove compounds from the stream, with the precipitants mostly depositing on benthic or macrophyte surfaces in the stream (Golterman, 2004; Parker et al., 2007; Tobias and Böhlke, 2011).

Additional references cited and added to manuscript:

Parker, S. R., Gammons, C. H., Poulson, S. R. and DeGrandpre, M. D.: Diel variations in stream chemistry and isotopic composition of dissolved inorganic carbon, upper Clark Fork River, Montana, USA, Applied Geochemistry, 22(7), 1329–1343, doi:10.1016/j.apgeochem.2007.02.007, 2007.

Tobias, C. and Böhlke, J. K.: Biological and geochemical controls on diel dissolved inorganic carbon cycling in a low-order agricultural stream: Implications for reach scales and beyond, Chemical Geology, 283(1), 18–30, doi:10.1016/j.chemgeo.2010.12.012, 2011.

*The authors use an extraction protocol (NaOH, BD, HCl) that is not very suitable to investigate authigenic Ca-P phases in the sediment. The HCl extraction removes large amounts of P that was supposedly associated with recalcitrant, geological apatite. I find the investigation of the possibility of Ca-P formation in the stream beds rather poorly developed with geochemical modelling of stream water and not interstitial water, and chemical extractions insuitable to explore authigenic Ca-P formation. The uncertainty around apatite exists throghout the paper.*

We see the referee's point and agree: had we used a P fractionation scheme designed to distinguish detrital/primary and authigenic Ca-P phases (e.g., the Ruttenberg (SEDEX) scheme), our P fractionation data would be suited to compare or contrast with the mineral equilibria results. However, no P fractionation scheme is perfect: with such a scheme, we would not have observed the forms of Fe oxides in the sediment which are quite likely to buffer DRP (amorphous Fe oxides vs crystalline oxides and recalcitrant phases). We chose another scheme because we implicitly thought Fe to be of more relevance to P attenuation in streams than Ca. We understand that the 'story' of the text did not necessarily give the right context for why we chose that fractionation scheme – we think our revisions mentioned already and below should clarify the matter.

However, we do not see this point (having not used a scheme suitable for authigenic Ca-P) as invalidating our conclusion that Ca-P precipitation is unlikely a control of DRP in streams of similar chemistry. More alkaline systems would necessitate more detailed investigation on Ca-P (e.g., karst). We just did not know a priori how alkaline the stream would have to be before the *potential* to form – which we measured – would be there.

Hence, we will address these concerns by 1) making our motivation for the Jan et al. 2015 scheme more apparent (make Fe a more central part of the story; see revised introduction above and further discussion below) and 2) acknowledging that, while the water-column mineral equilibria suggest little to no opportunity for Ca-P phases to affect DRP, a scheme such as SEDEX would help complement the observation and could be used in future work.

A sentence was added to the methods section 2.5, following the first sentence on L139:

While other schemes exist that provide alternative information on sediment P chemistry (Condron and Newman, 2011), we used the scheme of Jan et al. (2015) as it distinguishes between amorphous, reactive Fe oxides and more crystalline or recalcitrant Fe phases.

The text following the sentence on L306 has been revised to:

It is striking that some streams approach, but do not significantly extend beyond, the hydroxylapatite solubility curve, which could suggest a role for hydroxylapatite equilibrium. Future work could compare such stream chemistry data to sediment P fractionation data that includes detrital and authigenic Ca-P phases (e.g., Ruttenberg, 1992).

Additional references cited and added to manuscript:

Ruttenberg, K. C.: Development of a sequential extraction method for different forms of phosphorus in marine sediments, Limnology and Oceanography, 37(7), 1460–1482, doi:10.4319/lo.1992.37.7.1460, 1992.

*In the abstract, the authors mention that apatite does not occur in those streams. Then, in L304 it is mentioned that Ca-P formation is a best a secondary mechanism for P removal. Those are quite contrasting statements, and illustrate for me the unclear and unsatisfactory way in which Ca-P precipitation is deal with in the paper.*

We apologize for the confusion – our choices of words such as 'occur' were poor in hindsight.

What we meant in the abstract is that Ca-P minerals such as hydroylapatite or calcite were not really able to *form* and hence could not influence DRP in the water-column. We soften this notion in the discussion near L304 by pointing out that, technically, some waters were near saturation with respect to hydroxylapatite but, in practice, there's no real potential for that mineral to form (based on the literature cited in L303-304).

We will revise the text so that our language is more precise.

The sentence beginning in L13 is revised to "Neither P-containing minerals (e.g., hydroxylapatite) nor calcite-phosphate co-precipitation had the potential to form."

The sentence in L303-305 is revised to "Therefore, hydroxylapatite and other phosphate-mineral precipitation seems an unlikely mechanism for P removal in these streams."

*The main finding of the paper then appears to be that Fe (oxyhydr)oxides play an important role in P retention in these streams, and that reactive Fe abundance is a function of geology of the stream bed. In L362 and further, the authors then introduce literature from the field of Fe(II) oxidation and the resulting formation of a range of P-bearing Fe species (citation of the work of Senn et al. 2015), which seems quite out of context in this story as Fe redox cycling seems not to be a part of it. If it is, then the hyporeic zone is a very different geochemical zone altogether and using the stream water to explore mineral formaion makes little sense... To me, this is all a bit symptomatic of how bits and pieces of geochemistry are used in hypothetical and sometimes inappropriate ways, while there is relativey little robust data (solid-phase Fe and P fractionation, geochemical modelling of the stream water) to really pinpoint the processes governing P cycling and retention in the stream beds. The fact that poorly crystalline Fe (oxyhydr)oxides are important in P retention is very well-established, and therefore I have my doubts whether the findings in this contribution provide enough novel insight.*

The relationship between the amorphous Fe oxides and sediment P sorption is perhaps one of the most interesting findings in our opinion. It drives at several important points which we have tried to discuss in text. In addition to revisions already mentioned, here we'll discuss these points more directly.

1) Because of how well the BD-I Fe fraction is defined based on kinetic dissolution (see the Postma 1993 citation), this Fe fraction isolates very well the Fe oxides most responsible for P sorption (compare with other Fe extractions still in use such as longer-time dithionite (BD or CBD), oxalate, HCl, ascorbic acid, etc. – these are all much less specific than BD-I) and so enabled the clear relationship seen in Figure 7.

2) What the referee contends (*that poorly crystalline Fe (oxyhydr)oxides are important in P retention is very well-established*) is true in a general sense, but we argue that it's not true in the context of streams and P cycling (much stream literature is vague on this point). The specifics of the reactive surfaces responsible for sediment P sorption are often overlooked, *particularly in streams*. Stream studies often examine whole-stream P uptake, use various bioassays, or do simple sediment P sorption measurements (e.g., $EPC_0$) if sediments are considered part of the story at all. Select few stream studies consider the Fe oxides on sediment surfaces involved in sequestering P in streams – even fewer use an appropriate method to identify these Fe oxides. Much of the work where such robust data are collected are in terrestrial or limnetic systems (see references in text such as Marton and Roberts 2014 and Herndon et al. 2019). Further, the theme of how catchment lithology sets a template for the chemistry affecting P in streams is important but not often addressed thoroughly in the literature.

3) We suspect our second point above is the case because of the current stage of research on how the hyporheic zone attenuates P. If sediments play a role in P attenuation, then the hyporheic zone must be involved. Yet, the most encompassing and contemporary review on the hyporheic zone (Boano et al. 2014) itself mentions the paucity of research into P and the hyporheic zone whereas much more is known about nitrogen and other elements. We feel that the complexity involved (the surface chemistry around P sorption, competing interactions, the fact that sorption sites themselves (Fe oxides) are unstable and dynamic, etc.) and analytical constraints inhibit progress on this important topic. Our study is providing the necessary background so that more pointed research questions can arise: how exactly does the Fe cycle in streams work in regards to trapping/releasing P? Where exactly in the hyporheic zone (and how) do these amorphous Fe oxides arise? Are these Fe oxides steady through time under, say, baseflow? How do Fe colloids fit into the picture?

We think current understanding of P attenuation in streams is poor (referring to DRP simulations/models in the literature which struggle to replicate DRP time-series). It's our hope that studies like ours can begin to refine such efforts.

As discussed above, we see now that the theme of Fe cycling was not adequately developed before it jumps into the last two thirds of the discussion. We have revised the text (see earlier) to work this topic into the discourse more consistently.

*The authors mention that this is the first time that the extraction potocol of Jan et al. (2015) is used, but that novelty is of limited relevance to me, especially as I do not think much understanding on P burial in the stream beds is gleaned from it.*

We do not think 'burial' to be the case for streams, hence why we have repeatedly used a term like 'attenuation'; 'buffer'
would also be more apt than 'burial'. Even the sediments themselves have a finite residence time in streams. Much stream phosphorus work postulates the idea of benthic sediments buffering P (both adsorbing and desorbing P) but the workings behind that idea are often vague (once sorbed, P does not easily part with the sediment surface, as is known thoroughly from the soil sciences). Fe cycling, as we discuss, could be one way that the stream at baseflow can maintain P concentrations that are high enough to promote eutrophication. For example, perhaps in some areas of the stream, Fe oxides effectively trap P
but elsewhere those Fe oxides are being reductively dissolved and slipping past zones of re-oxidation. Our study provides some context on this and it would not have been possible without the two BD fractions in the Jan et al. 2015 scheme.

*Lastly, nothing is really done with the findings. As mentioned, in the introduction it does not become clear what relevance*
*the understanding of P retention mechanisms in the investigated streams has, and in the discussion and conclusion it does not become apparent what the implications of the findings are. The findings fit with well-established concepts of Fe-controlled P mobility, but there is no link to the situation/future/management of the investigated streams.*

*Overall, this reviewer is left wondering whether the identification of a well-established P retention mechanism in New*
*Zealand streams, while the finer details of P burial are obscured because of limited data and dependence on uncertain geochemical modelling, warrants publication in BG. Especially as neither the context for studying this area, nor the implications of the findings, are very well-developed.*

We touch on the primary relevance of the work in the opening paragraph of the introduction. The problem is not so much in
knowing how streams retain P *but also how they might release it back into the water-column.* This necessitates knowing the mechanisms responsible.

We agree that the relevance of the work seems to not neatly tie back into the 'big problem': that of legacy P. Hence, we have revised parts of section 4.4 to better incorporate the theme given in the introduction.
A sentence in the first paragraph (starting L384) was revised (original sentence starting at L391):

Both of these potential mechanisms for a slow release of sediment P into the water-column (thus increasing or maintaining DRP) are constrained for streams with little hyporheic exchange (e.g., the spring-fed streams in the present study; Boano et
al., 2014).

Further, the second paragraph of section 4.4 (L395) was revised:

Further research is needed to link the abiotic P exchange mechanisms we have discussed to the spatiotemporal DRP signal
observed in the water column. Reach-scale studies of P attenuation (Ensign and Doyle, 2006; Hall et al., 2013) are unspecific regarding how streams control DRP concentrations and therefore buffer P pollution within the catchment. While stream sediments are well-known to provide reaction sites for P, little has been done to link these reaction sites in streams to P cycling in a mechanistic manner, i.e., by connecting these zones of reactivity to hydrological transport (Boano et al., 2014). We suggest that future research on stream P cycling avoids mono-causal interpretations (Kalbitz et al., 2017), and incorporates the multiple mechanisms – both biotic and abiotic – where relevant. With detailed understanding, we may be able to address the problem of legacy P and how to better mitigate P pollution of our waters.

We hope that by focusing the story in the paper, we have made our contribution more apparent and of significance to the community focused on tackling watershed P problems.

---

## Author Comment (AC2) · 18 Mar 2020

We have taken the opportunity to further address the comments raised in referee #2's report, which are summarized as follows: 1.The fact that the geochemical modelling was done in the water and not in the sediment interstitial water where the geochemistry will be very different and the key P reactions are likely to be taking place. The reliability of the modelling is also further in question because of the presence of colloidal Fe-P phases that pass through the 0.45 micron filters. 2. The extraction scheme used does not identify Ca associated P which is highly relevant to the modelling 3. The key finding that P sorption is most closely associated with amorphous Fe is not really new

or insightful in this case.

In this process, we have refocused the paper via major revisions and so do not go into detailed changes to the text.

Broadly, we have more carefully treated the topic of Ca cycling in streams and have included new data (an estimate of authigenic Ca-P phases in sediments) to support our discussion, following concern raised in referee #2's report. We balance the themes of Ca and Fe cycling more carefully and make Fe cycling a more apparent theme. Most importantly, we more clearly identify the novel finding of our study: that despite being much more sorptive due to greater poorly crystalline Fe oxide content, streams with sediments with high P sorption capacity actually had greater (not less) DRP. Overall, we think this discussion has forced us to sharpen our points made in the paper and think the paper has greatly benefited as a result.

1.The fact that the geochemical modelling was done in the water and not in the sediment interstitial water where the geochemistry will be very different and the key P reactions are likely to be taking place. The reliability of the modelling is also further in question because of the presence of colloidal Fe-P phases that pass through the 0.45 micron filters.

We make effort to constrain our discussion to the quick hyporheic flow paths (very upper few cm of the benthic substrate). While deeper hyporheic waters – where geochemical and nutrient changes relative to the water column would be most pronounced – are generally important, they require more difficult sampling designs and would target much slower flow paths (i.e., low rates of exchange with the water column).

Having said that, we do note more clearly how previous research has shown, for example, calcite deposition/dissolution in streams: such mineral formation creates deposits on the upper benthic substrate (gravel, cobbles, leaves, etc.) but may be incorporated into interstitial sediments over time.

While the likely presence of Fe colloids which passed our conventional 0.45 micron filters does bias equilibria regarding Fe phases, this bias does not affect Ca or other mineral equilibria. As a check, we ran a sensitivity analysis (data not shown) where input Fe concentrations were cut in half before entered into PHREEQC: only Fe phase results changed (e.g., ferrihydrite SI's) while others (e.g., calcite, hydroxylapatite) remained unaffected.

Related to this point, we moved much of the discussion regarding geochemical equilibria out of the main text and into the supplementary material. This has helped to focus the results and outcomes of the study.

2. The extraction scheme used does not identify Ca associated P which is highly relevant to the modelling

We initially chose the Jan et al 2015 scheme because of our interest in Fe oxides and P sorption, while the focus of the study does lean more towards Fe now, we agreed that this point is relevant to the discussion around Ca-P cycling.

Hence, we took the opportunity to measure an authigenic Ca-P phase as suggested by referee #2. Using freeze dried sediments we had stored, we applied a modified SEDEX procedure to generate this P fraction (termed acetate-P, inline with how the other P fractions are termed according to their extractant). Additionally, we measured the following HCl-P step: the sum of acetate-P and this SEDEX HCl-P agreed very well with the original HCl-P data (see attached figure).

The acetate-P fraction is discussed in detail in the text. Notably, we think this fraction overestimated authigienic Ca-P, due to organic matter content (which has been observed by others cited in text). But, the fraction did appear to relate to P sorption capacity (along with Fe oxides), possibly related to carbonates (like calcite) providing some reactivity. Unfortunately, few methods exist currently to examine authigenic Ca-P phases in stream sediments.

3. The key finding that P sorption is most closely associated with amorphous Fe is not really new or insightful in this case.

We had not fully reasoned through our dataset which made this point the most important to address. Of course, it's thoroughly documented that poorly crystalline Fe oxides are strongly P sorptive and we do not wish to make this a key finding. Rather, we revised the manuscript to focus on the unexpected finding that, for streams with permeable sediments, greater ASC (P sorption capacity) did not translate into lower DRP concentrations but instead was associated with increases in DRP. We think this is a critical observation, which forces us to think more deeply about the (bio)geochemical cycling of Fe in streams. We have reframed our discussion regarding this key finding and think the story is clearer and more effective for it.

We thank referee #2 for these comments.

[Figure]

**Fig. 1.**